



# Empirically-Derived Parameterizations of the Direct Aerosol Radiative Effect based on ORACLES Aircraft Observations

Sabrina P. Cochrane[1,2], K. Sebastian Schmidt[1,2], Hong Chen[1,2], Peter Pilewskie[1,2], Scott Kittelman[1], Jens Redemann[3], Samuel LeBlanc[4,5], Kristina Pistone[4,5], Meloë Kacenelenbogen[4], Michal Segal
Rozenhaimer[4,5,6], Yohei Shinozuka[4,7], Connor Flynn[8], Amie Dobracki[9], Paquita Zuidema[9], Steven Howell[10], Steffen Freitag[10], Sarah Doherty[11]

[1]Department of Atmospheric and Oceanic Sciences, University of Colorado, Boulder, 80303, USA
[2]University of Colorado, Laboratory for Atmospheric and Space Physics, Boulder, 80303, USA
[3]School of Meteorology, University of Oklahoma, Norman, Oklahoma, 73019, USA
[4]NASA Ames Research Center, Mountain View, 94035, USA
[5]Bay Area Environmental Research Institute, Mountain View, 94035, USA
[6]Department of Geophysics and Planetary Sciences, Porter School of the Environment and Earth Sciences, Tel-Aviv University, Tel-Aviv, Israel
[7]Universities Space Research Association/NASA Ames Research Center, Mountain View, 94035, USA
[8]Pacific Northwest National Laboratory, Richland, Washington, 99354, USA
[9]Department of Atmospheric Science, Rosenstiel School of Marine and Atmospheric Science, University of Miami, Miami, FL, 33146, USA
[10]Department of Oceanography, University of Hawaii, Honolulu, HI, 96822, USA
[11]Joint Institute for the Study of Atmosphere and Ocean, University of Washington, Seattle, WA, 98195, USA

*Correspondence to*: Sabrina P. Cochrane (sabrina.cochrane@colorado.edu)

**Abstract.** This work establishes an observationally-driven link from mid-visible aerosol optical depth (AOD) and other scene parameters to broadband shortwave irradiance (and by extension, the direct aerosol radiative effect, DARE), based on observations from the 2016 and 2017 field campaigns of ORACLES (ObseRvations of Aerosols above CLouds and their intEractionS). Specifically, this is done by two parameterizations, one spanned by the mid-visible AOD and scene albedo
below the aerosol layer, and another one with a third input, the mid-visible aerosol single scattering albedo (SSA). These parameterizations build on the earlier concept of radiative forcing efficiency, which describes the dependence of DARE on the AOD, and extend it to make the dependence on the other two scene parameters explicit.

The parameterizations are founded on 9 cases from the campaigns, for which we retrieve the spectral aerosol properties of SSA and asymmetry parameter (*g*) directly from the radiative fluxes, based on the method presented in Cochrane et al., 2019.
These properties are used as the basis of the parameterizations, capturing the natural variability of the study region as sampled. The majority of the case-to-case variability within the ORACLES DARE dataset is attributable to the dependence on AOD and scene albedo. This is captured by the first parameterization, which is advantageous when satellite retrievals provide only limited information such as AOD and scene albedo. However, the second parameterization explains even more of the case-to-case variability by introducing the mid-visible SSA as third parameter. For both parameterizations, we provide the necessary
coefficients, uncertainties, and code required for the user to reconstruct the parameterization for their use.



## 1 Introduction

During the African burning season of August-October, a semi-permanent stratocumulus cloud deck just off the southern African west coast is overlaid by a thick layer of biomass burning aerosols. These aerosols are advected over the southeast Atlantic Ocean from the interior of the African continent and account for nearly 1/3 of the total global biomass burning aerosol (van der Werf et al., 2010). The seasonal environment of high biomass aerosol loading above clouds has large, variable radiative impacts that have yet to be fully characterized.

In addition to many other science objectives, the NASA ORACLES aircraft campaign aimed to obtain the Direct Aerosol Radiative Effect (DARE) in both cloudy and clear skies for this region (Zuidema et al., 2016; Redemann et al., in prep.). The distinction between DARE in cloudy versus clear skies is crucial since the albedo below an aerosol layer strongly influences the sign and magnitude of DARE. The albedo from below an aerosol layer can determine the sign of the top of the atmosphere (TOA) DARE independently of the aerosol itself (Twomey, 1977; Hansen et al., 1997; Russell et al., 2002; Keil and Haywood, 2003; Yu et al., 2006; Chand et al., 2009; Zhang et al., 2016; Meyer et al., 2013; Meyer et al., 2015). In a region like the southeast Atlantic, this makes determining DARE challenging since the cloud fields change rapidly. Depending on the cloud albedo, the aerosol could be warming (positive DARE) or cooling (negative DARE) at the TOA (Yu et al., 2006; Russell et al., 2002; Twomey, 1977). The albedo value where DARE transitions from positive to negative, or warming to cooling, is known as the critical albedo (Chand et al., 2009).

The spectral DARE in Wm$^{-2}$nm$^{-1}$ is determined from the difference between the net irradiance ($F_\lambda^{net}$) with and without the aerosol layer:

$$DARE_\lambda = F_{\lambda,aer}^{net} - F_{\lambda,no\ aer}^{net}. \tag{1}$$

Aircraft measurements, such as those taken during ORACLES, provide direct observations of the components necessary to calculate DARE. However, measurements are only taken for a sample of the region and may not be representative of the region as a whole. DARE calculated from aircraft observations alone would therefore leave the larger question of whether the aerosols warm or cool the southeast Atlantic unanswered.

In the case of DARE, the translation from individual observations into a common framework was first introduced by Meywerk & Ramanathan (1999). The radiative forcing efficiency (RFE) empirically relates DARE to the aerosol optical depth (AOD):

$$DARE=RFE*AOD. \tag{2}$$

The RFE is defined as the (usually broadband) DARE normalized by the (usually mid-visible) AOD, or sometimes as the derivative of DARE with respect to the AOD. It can be regarded as an intensive property of an airmass that allows the direct conversion from AOD to DARE, complementing calculations based on aerosol microphysical and optical properties. When



the RFE is aggregated for an entire field mission, it can provide a mean airmass characteristic that lends aircraft observations a broader scientific impact than the contributing individual measurements. If aerosol microphysical and optical properties are insufficiently known in a region of interest, this mission-aggregated RFE can carry a DARE parameterization that solely requires AOD (equation 2). If the RFE varies little in a region and season of interest, it can be used to derive regional DARE estimates via AOD statistics from satellites – at least in principle. More fundamentally, observations of the dependence of flux

changes on aerosol optical depth help to develop confidence in radiative forcing calculations based on measured aerosol properties (Russell et al., 1999, Redemann et al., 2006). In this sense, the RFE in conjunction with Eq. (2) provides closure to those calculations, and thus constrains those from the radiative flux and DARE perspective.

In this paper, we generalize the concept of RFE by explicitly taking into account the dependencies of DARE not only on AOD as expressed in equation (2), but also on both the aerosol and cloud properties. ORACLES measurements are used collectively

to develop two parameterizations in the form of:

$$DARE = P(AOD_{550\ nm},\ \alpha_{550\ nm}), \tag{3}$$

and

$$DARE = PX(AOD_{550\ nm}, \alpha_{550\ nm}, SSA_{550\ nm}), \tag{4}$$


where AOD, $\alpha$ and SSA are the aerosol optical depth, albedo, and single scattering albedo at 550 nm. The 550 nm albedo is the albedo of the scene below the aerosol layer, and the SSA is a measure of aerosol absorption. *P* stands for the two-parameter representation of DARE and *PX* stands for an extended version with three parameters. Both parameterizations provide instantaneous *broadband* DARE that are based upon *spectral* aerosol and cloud properties. In other words, the right-hand sides

of equations 3 and 4 are understood to be narrowband quantities, while the left-hand sides are the broadband results. The parameterizations have the advantage of implicitly accounting for the spectral dependencies of the aerosol and cloud properties (e.g. aerosol scattering phase function, aerosol vertical distribution, spectral dependence of aerosol absorption, cloud optical depth, cloud effective radius, cloud top and base height), whereas the dependence on mid-visible AOD, SSA, scene albedo, and solar zenith angles is explicit.

From the user standpoint, applying the parameterizations is straightforward because broadband DARE can be estimated with minimal information on the cloud and aerosol properties. The parameterization coefficients encompass the many complexities of transitioning from narrowband to broadband, such that the spectral dependencies of the cloud and aerosol properties are not necessary. Of course, the parameterization only represents the "mean" conditions encountered in the ORACLES region and sampling time, and it becomes invalid outside of this mission envelope. Equation (3) only requires AOD and scene albedo at

mid-visible 550 nm, which can be readily obtained from satellite observations. If mid-visible SSA is also known (from satellite or aircraft retrievals, in-situ observations, or from a climatology), the second parameterization (Eq. 4) can be used, which decreases the uncertainty of DARE, as we will discuss below.



To arrive at the final parameterizations, we first build upon the method presented in Cochrane et al. (2019) and determine the aerosol intensive properties of SSA and asymmetry parameter (*g*) that best represent the ORACLES region during August and

September of 2016 and 2017. We evaluate the radiative effects of those aerosols where the relationships found between DARE, AOD, and albedo form the foundation of the parameterizations that capture the collective variability sampled from the viable cases from ORACLES 2016 and 2017.

In Section 2, we describe the data and the methods used to determine spectrally resolved SSA and *g* and how they are used within the DARE calculations to construct the parameterizations. Section 4 provides results and discussion of aerosol retrievals

and the parameterizations for the relationship between radiative effects, AOD, albedo, and SSA. In Section 5, we provide a summary and discussion of the results.

## 2 Data and Methods

### 2.1 Data

The ORACLES project conducted research flights in the southeast Atlantic for 3 one-month periods over three consecutive

years (2016-2018) during the burning season to study the biomass burning aerosols and stratocumulus cloud deck. To achieve the defined science objectives, the ORACLES project made use of the NASA P-3 aircraft for the duration of the experiment and the NASA ER-2 aircraft in 2016 only. Between the 2016 and 2017 deployments, the P-3 completed 26 science flights, five of which were collocated with the ER-2. All data can be found on the NASA ESPO archive website (ORACLES Science Team, 2017a, b, 2019).

We focus on utilizing measurements taken from the P-3, primarily the irradiance measurements taken by the Solar Spectral Flux Radiometer (SSFR, Pilewskie et al., 2003; Schmidt and Pilewskie, 2012) in conjunction with AOD and retrievals of column gas properties from the Spectrometer for Sky-Scanning Sun-tracking Atmospheric Research (4STAR, Dunagan et al., 2013; Shinozuka et al., 2013; LeBlanc et al., 2020) to achieve the specific goals of this paper. SSFR consists of two pairs of spectrometers. Each pair (one zenith viewing and one nadir viewing) covers a wavelength range of 350-2100 nm. SSFR is

radiometrically and angularly calibrated pre- and post- mission. Its zenith light collector is equipped with an active leveling platform (ALP), which keeps it horizontally aligned by counteracting the variable aircraft attitude. This allows the collection of irradiance data as long as pitch and roll stay within the ALP operating range of 6°. This ensures that radiation from the lower hemisphere does not contaminate the zenith irradiance measurements, which was especially important for the bright clouds encountered during ORACLES. 4STAR provides spectral retrievals of AOD from the solar direct beam irradiance

above the aircraft and is calibrated through Langley extrapolation technique before and after deployment at Mauna Loa Observatory along with in-flight high-altitude measurements (see LeBlanc et al., 2020 for details on 4STAR calibration). 4STAR also provides aerosol intensive properties (e.g., SSA described in Pistone et al. 2019), column water vapor and trace gas retrievals, such as ozone (e.g., Segal-Rosenheimer et al., 2014). Further details on SSFR, ALP and 4STAR instrumentations and calibrations can be found in Cochrane et al., 2019.



## 2.2 Methods

To construct our DARE parameterizations, aerosol intensive optical properties such as SSA and *g* must be determined for as
many cases as possible. Retrieving these properties from aircraft irradiance measurements is inherently challenging because
the aerosol radiative effects can be relatively small compared to the horizontal variability of cloud albedo.
Cochrane et al., 2019 showed for two cases that special spiral maneuvers ("square" spiral) are more successful than the heritage
"stacked leg" approach because multiple measurements are taken throughout the vertical profile over a short time period
(typically 20 minutes). This sampling strategy reduces the effects of cloud inhomogeneities and allows isolation of the aerosol
signal, as long as specific quality criteria (detailed below) are met. These criteria, preceded by two filtering steps in which data
points are removed, are described in the following section and follow the order presented in the flow chart of Figure 1. The
filters and criteria provide objective data conditioning prior to the subsequent aerosol retrieval and DARE parameterizations.

### 2.2.1 Data Conditioning

Throughout the spiral, the zenith (downwelling) and nadir (upwelling) irradiance measurements are continuously affected by
the aerosol layer. The aerosol-induced changes to the irradiance profiles allow us to extract information about the aerosol itself.
As can be seen in Figure 3a, both upwelling ($F_\lambda^\uparrow$) and downwelling ($F_\lambda^\downarrow$) irradiance profiles have an approximately linear
relationship to AOD due to the absorption and scattering of the aerosol layer. Any deviation from the linear relationship is
attributed to changes in the underlying cloud; these are filtered out to isolate the radiative effect of the aerosol.
Following the methods described in Cochrane et al. (2019), two filters are applied to the data to ensure the isolation of aerosol
effects. Prior to filtering, all data are corrected to the SZA at the midpoint of the spiral according to Equation 3 in Cochrane et
al. (2019) to account for the minor change in solar position throughout the spiral. The first is an altitude filter (see F1 in Fig.
1), where the altitude range is limited to encompass only the vertical extent of the aerosol layer. The second is a homogeneity
filter (see F2 in Fig. 1), which selects the dominant profile of measurements, whether that be cloudy or clear sky, and removes
any outlying data. The filter begins with a linear fit of the irradiances with respect to the AOD for each wavelength:

$$F_\lambda^\uparrow = a_\lambda^\uparrow + b_\lambda^\uparrow * AOD_\lambda, \tag{5}$$
$$F_\lambda^\downarrow = a_\lambda^\downarrow + b_\lambda^\downarrow * AOD_\lambda, \tag{6}$$

where $a_\lambda$ and $b_\lambda$ are the slope and intercept of the linear regression, for which the individual data points are weighted inversely
by the irradiance uncertainties. In any particular spiral, the measurements could be taken from either predominantly cloudy or
clear sky. The filter, which is applied to the upwelling profile, retains only those data within the 68% confidence interval (1
sigma) of the linear fit line. This ensures that the retained data contains no outlying points and is all from one mode: clear sky
or cloudy sky. This filtering step is slightly modified from the method presented in Cochrane et al. (2019) in two ways: 1) the
irradiances were previously fit against AOD at 532 nm only rather than AOD at the corresponding wavelength and 2) the range



of retained data was previously based on the confidence interval of overall mean irradiance value rather than the confidence interval of the linear fit throughout the profile. We have made these adjustments to better allow for linear variation with altitude while eliminating data that significantly deviates from the profile. There are 2 exception cases for which we maintain the

original filtering from Cochrane et al., 2019 using the confidence interval on the mean value. For these cases, the filtering modification overly eliminated data or retained excessive variability at small AOD values (high altitude).

Following the filters, each case must pass criteria that ensure the changes in net irradiance with altitude are caused by the aerosol radiative effects and not variability in the underlying cloud field. First, irradiance measurements must be available throughout the spiral, spanning the full AOD dynamic range between the top and bottom of the layer (C1 in Fig. 1) The most

common reason for cases to fail this criterion is that the AOD never reaches background stratospheric AOD levels (near zero; 0.02-0.04 in the midvisible), indicating measurements were not taken fully above the aerosol layer. Since the retrieval relies on the change in irradiance with altitude, incomplete profiles do not provide a sufficient change required to capture the aerosol signal.

The second requirement (C2 in Fig. 1) is to ensure that the true aerosol absorption be larger than the 3-D cloud effect known

as horizontal flux divergence (see Fig 1 in Cochrane et al., 2019). SSFR actually does not measure the absorption directly, but rather the decrease of the net flux $F_\lambda^{net}$ from the top of the aerosol layer (TOL) to the bottom (BOL), or vertical flux divergence:

$$V_\lambda = \frac{(F_{\lambda,tol}^{net} - F_{\lambda,bol}^{net})}{F_{\lambda,tol}^{\downarrow}} = \frac{[(F_{\lambda,tol}^{\downarrow} - F_{\lambda,tol}^{\uparrow}) - (F_{\lambda,bol}^{\downarrow} - F_{\lambda,bol}^{\uparrow})]}{F_{\lambda,tol}^{\downarrow}},$$ (7)

which we normalized by the incident irradiance. $V_\lambda$ is only the *vertical* part of the total flux divergence. The other part is the *horizontal* flux divergence, $H_\lambda$, which is not measured by SSFR. The true absorption, $A_\lambda$, is obtained from the *total* flux divergence:

$$A_\lambda = V_\lambda - H_\lambda.$$ (8)


If the condition $|H_\lambda| << |V_\lambda|$ (see section 3.1.2 in Cochrane et al., 2019), then $A_\lambda \approx V_\lambda$, and the vertical flux divergence measured by SSFR can be used in lieu of the true absorption. The first step to check that this requirement is met is to calculate $V_\lambda$ from the linear fit in equations (5) and (6):

$$V_\lambda = \frac{AOD_{532}^{max} * (b_\lambda^{\uparrow} - b_\lambda^{\downarrow})}{a_\lambda^{\downarrow}} \ ,$$ (9)

where $AOD_{532}^{max}$ is the AOD at the bottom of the spiral (just above the cloud), and $a_\lambda$, $b_\lambda$ are the slope and intercept of the linear fit lines. The second step is to estimate $H_\lambda$. Neglecting its weak wavelength dependence (Song et al., 2016), we instead use



$H_\infty$, the value of $H_\lambda$ at large wavelengths. As described in Cochrane et al. (2019), $H_\infty$ can be determined using measurements

of $AOD_\lambda$ and $V_\lambda$: the aerosol optical depth decreases with increasing wavelength, and therefore the true aerosol absorption decreases as well; as $AOD_\lambda$ reaches zero, so does $A_\lambda$. When this happens, any non-zero measured value of $V_\lambda$ must originate from $H_\lambda$ because $A_\lambda = 0 = V_\lambda + H_\lambda$. Since this occurs at long wavelengths, the vertical flux divergence $V_{\lambda \to \infty}$ yields $H_\infty$. In practice, we obtain $H_\infty$ from the intercept of the regression between $AOD_\lambda$ and $V_\lambda$.

To determine the relative amount of absorption to horizontal flux divergence, Cochrane et al. (2019) developed a unitless

metric ($i_\lambda$) that determines whether the case is viable for an aerosol retrieval. $i_\lambda$ is defined as:

$$i_\lambda = \frac{H_\infty}{V_\lambda - H_\infty},\qquad(10)$$

If $i_\lambda > 0.3$, then the condition $|H_\lambda| << |V_\lambda|$ is not met, and the case is not considered viable for a subsequent retrieval.

The final criteria (C3 in Fig 1.), the measured albedo at the cloud top (Bottom of Layer, BOL) and above the aerosol layer (Top of Layer, TOL) shown in 3b must be consistent in the limit of zero AOD. As the aerosol absorption decreases with increasing wavelength, the ratio between the measured albedo at the cloud top (BOL) and above the aerosol layer (TOL) must shift closer and closer to 1. Analogous to the determination of $H_\infty$ and illustrated in Figure 2c, we determine $AR_\infty$ as the intercept between the TOL and BOL albedo ratio and the AOD:

In the limit of: $\lambda \to \infty$:

$$\lim_{AOD(\lambda) \to 0} \frac{albedo_{\lambda, TOL}}{albedo_{\lambda, BOL_\lambda}} \equiv AR_\infty.\qquad(11)$$

$AR_\infty$ is our final criterion, and any deviation larger than 0.1 from 1.0 (i.e., the intercept must fall between 0.9 and 1.1) indicates

that other factors affect the data besides the aerosol absorption. For example, a changing cloud field could change the albedo between the beginning and end of the spiral, and the aerosol retrieval might wrongly attribute this change to aerosol absorption. To summarize, the criteria each case must pass are:

C1. There must be valid data from both SSFR and 4STAR throughout the entire aerosol profile. Cases cannot be used within the retrieval if there is a lack of data due to aircraft flight pattern, ALP malfunction, or AOD data flagged for bad quality.

C2. $|i_\lambda|$ must be below 0.3 to ensure that the aerosol absorption is large enough compared to the horizontal flux divergence so that an aerosol retrieval is possible.

C3. $AR_\infty$ must fall between 0.9 and 1.1 to ensure that the spectral albedo is consistent both above and below the aerosol layer. Both the filters and the criteria are designed to control for any rapidly changing, potentially inhomogeneous cloud field encountered during ORACLES. Table 1 presents the C2 and C3 criteria and retrieval status of $SSA_\lambda$ and $g_\lambda$ for spiral cases

completed in 2016 and 2017 that passed C1. The criteria for which a case fails is indicated in red text. In 2016, five spiral


profiles out of 18 met all criteria, while four out of 23 met the criteria in 2017. Table 2 provides the UTC, latitude, and longitude ranges for each successful spiral profile.

### 2.2.2 Retrieval Algorithm

If a spiral irradiance profile has passed every criteria metric, the aerosol property retrieval is run. The retrieval, described in
detail in Cochrane et al. (2019), is based on statistical probabilities between the calculated model irradiance profiles and the measured irradiance profiles. The retrieval process is similar to curve-fitting, where we vary the parameters in question (i.e. SSA and $g$) until the radiative transfer model (RTM) calculations best fit the measured data.

The SSA and $g$ retrieval is performed with the publicly available 1-dimensional (1D) RTM DISORT 2.0 (Stamnes et al., 2000) with SBDART for atmospheric molecular absorption (Ricchiazzi et al., 1998) within the libRadtran library (Emde et al., 2016;
libradtran.org). For each wavelength, we use the RTM to progress through pairs of SSA and $g$ and calculate the upwelling, downwelling, and net irradiance profiles for each pair. For each {SSA, $g$} pair calculation, a probability is assigned to every SSFR data point in the profile according to the difference between the calculation and the measurement based on an assumed Gaussian distribution that represents the SSFR measurement uncertainty. The overall probability of a specific {SSA, $g$} pair given the SSFR irradiance measurements is the product of the individual probabilities for each data point; the {SSA, $g$} pair
with the highest overall probability between all three profiles (upwelling, downwelling, net) is the retrieval result for that wavelength. The inclusion of the net profile is an expansion upon the method described in Cochrane et al. (2019). The net irradiances provide a direct absorption constraint on the SSA retrieval, whereas the asymmetry parameter retrieval draws primarily upon the upwelling and downwelling fluxes.

In addition to the aerosol property pairs of {SSA, $g$}, the RTM ingests the spectral cloud top albedo from SSFR and the aerosol
extinction profile derived from the 4STAR AOD profile. The AOD profile has been conditioned such that the profile decreases monotonically to eliminate any unphysical extinction values (i.e., negative extinction). Any remaining AOD above the aerosol layer is allocated to a layer extending to an altitude of 15,000 m.

We modified the standard tropical atmosphere included in the libRadtran package (Andersen et al., 1986) to include the column water vapor measurements taken by the NASA P3 hygrometer from the level of the cloud top to the maximum altitude of the
spiral; the values at altitudes that are not informed by aircraft measurements are set to the standard tropical atmosphere values. The full water vapor column was then scaled to the water vapor value retrieved with 4STAR. The column ozone amount in the standard tropical atmosphere is also scaled by the column ozone amount retrieved with 4STAR. The solar zenith angle (SZA) is set to the mean SZA of the spiral. Table 2 lists, for each spiral case, the UTC, latitude, longitude albedo at 500 nm, mean SZA, AOD at 500 nm, column water vapor, and column ozone.

For 4 cases, the retrieval is possible only for $SSA_\lambda$ and not for $g_\lambda$. This occurs when the irradiance profiles a) did not have enough data points and/or b) are subject to scene inhomogeneities despite the filters and criteria described in the previous section. The $g$ retrieval is less sensitive than the SSA retrieval since the effect of $g$ is smaller than that of SSA on the irradiance profile. For these specific cases, the retrieval is modified such that $g$ is an input to the retrieval rather than a variable, and SSA



is the only retrieved parameter. For each wavelength, the input of $g$ is set to the mean value from the cases for which we had

valid $g$ retrievals. Table 1 lists which properties (SSA and $g$; SSA only) were retrieved for each case.

## 2.3 DARE

### 2.3.1 DARE Calculations

The retrieved pairs of $SSA_\lambda$ and $g_\lambda$ serve as the aerosol properties for the $DARE_\lambda$ calculations that the parameterizations are based upon. $DARE_\lambda$ can be calculated at any level. We focus on the TOL calculations since radiative effects can be directly

related to radiative balance at the TOL (Matus et al., 2015).

For each pair of retrieved $SSA_\lambda$ and $g_\lambda$, we calculate instantaneous $DARE_\lambda$ for SZAs from 0° to 80° with a 10-degree resolution for a range of albedo and AOD values. Since the $SSA_\lambda$ and $g_\lambda$ retrievals are valid only for the shortwave wavelength range ($\lambda \leq 781$ nm), we extend to longer wavelengths (up to 2100 nm) as described in detail in Appendix A.

Finally, the albedo must be generalized to all SZAs for a range of albedo spectra to be used within the $DARE_\lambda$ calculations.

Since we measure albedo only at a single SZA, we must use the RTM to determine the spectral shape and magnitude of the albedo at each SZA. We make this transition via a cloud retrieval; cloud properties of effective radius and cloud optical thickness (COT) are retrieved from the original cloud top albedo spectrum measured by SSFR at the bottom of the spiral. The effective radius is then held constant and the albedo spectra are calculated for a range of COTs at each SZA. Specific details of the albedo calculations can be found in Appendix A.

At each SZA, the RTM is run twice for each set of AOD values and cloud albedo spectra: with and without the aerosol layer included. The difference between the two runs is $DARE_\lambda$. The calculations are completed for wavelengths between 350 and 2100 nm; the integration of the $DARE_\lambda$ spectrum provides broadband DARE. This is done for each pair of $SSA_\lambda$ and $g_\lambda$.

### 2.3.2 Parameterizations

In the past, the Radiative Forcing Efficiency served the purpose of scaling measurements to larger regions and into climate

models. However, the RFE excludes both the dependence of DARE on cloud albedo and the non-linearities of the DARE-AOD relationship. While the non-linearity of RFE has long been known (Russel et al., 1997; Forster et al., 2007; deGraaf et al., 2012), no studies that we are aware of have generalized RFE to account for these complexities in a quantitative framework. Our first goal was to develop a parameterization that builds upon the RFE concept and generalizes it to explicitly include the dependencies and non-linearities that the RFE excludes while maintaining simplicity. The parameterization ($P_{DARE}$) provides

a broadband DARE estimate with minimal inputs in the form:

$$DARE = P(AOD_{550}, \alpha_{550}) = L(\alpha_{550}) * AOD_{550} + Q(\alpha_{550}) * AOD_{550}^2 \qquad (12)$$





where $L$ and $Q$ are the parameterization coefficients and $\alpha_{550nm}$ and $AOD_{550nm}$ are required inputs of 550 nm albedo and 550

nm AOD, respectively. $P_{DARE}$ has the significant advantage that the complexities of transitioning from narrowband to broadband for many parameters are incorporated into the parameterization coefficients, allowing for use across large spatial scales since minimal information is required as input.

Our second goal was to increase the level of complexity of the $P_{DARE}$ parameterization by including the additional constraint of the aerosol SSA. While $P_{DARE}$ requires minimal input, the more advanced parameterization, $PX_{DARE}$, includes the 550 nm

SSA as an additional parameter; this decreases the variability between cases. $PX_{DARE}$ is in the form:

$$DARE = PX(AOD_{550}, \alpha_{550}, \Delta SSA_{550}) = P(AOD_{550}, \alpha_{550}) + \Delta(AOD_{550}, \alpha_{550}, \Delta SSA_{550}), \qquad (13)$$

where the first term on the right-hand side is $P_{DARE}$ (Equation 12) and the second term (delta term) represents the change in

DARE due to varying SSA.

The coefficients of $P_{DARE}$ and $PX_{DARE}$ are determined based on the DARE calculations performed for each case with the associated pair of $SSA_\lambda$ and $g_\lambda$, with the end result of two parameterizations that empirically represent the relationship between DARE and its driving parameters while capturing the variability between individual cases. Further details of the $P_{DARE}$ and $PX_{DARE}$ development are best understood in conjunction with result figures and explained in further detail in Section 3.2.

**3. Results and Discussion**

**3.1 Aerosol Properties**

Figure 3a shows the retrieved asymmetry parameter values for each case with sufficient sensitivity. The red dashed line represents the average spectrum, where the error bars are calculated by propagating the uncertainty of each individual retrieval (shown in Appendix E). The average spectrum is used in the SSA retrievals for cases that did not have sufficient sensitivity to

retrieve $g$.

The asymmetry parameter decreases with increasing wavelength more rapidly than found in AERONET retrievals from sites in the SE Atlantic (São Tomé, Ascension Island and Namibia; Appendix B, Fig B2). The AERONET retrieval algorithm is fundamentally different from the one used here. The AERONET operational inversion method assumes a size-independent complex refractive index (Dubovik and King, 2000), which can potentially lead to errors in the retrieved size distribution from

which the optical properties are determined (Dubovik et al., 2002; Dubovik et al., 2006; Chowdhary et al., 2001). At 550 nm, the average $g$ value is 0.52; by 660 nm, $g$ has dropped to 0.43. Simple Mie calculations, shown in Appendix B, confirm that this spectral dependence is possible with a particular fine to coarse mode aerosol ratio. In addition, the AERONET sites are located at the perimeter of the ORACLES study region: At the very north (São Tomé), west (Ascension) and southeast





(Namibia) ends of where the P3 flew. As such, the aerosol measured at the AERONET sites might actually differ from that
measured during our retrievals.

Figure 3b shows the retrieved SSA spectra from each successful spiral case, and the mean retrieved SSA and *g* for each
wavelength are presented in Table 3. Our retrievals of SSA range from 0.78 to 0.88 at 550 nm, with an average value of 0.83.
The red spectrum shows the mean of all cases. The SSA retrieved through our new method is spectrally flatter than reported
from the SAFARI 2000 campaign, which took place in the southeast region of the ORACLES measurement domain (Eck et
al., 2003; Haywood et al., 2003; Russel et al., 2010). The SAFARI SSA values tend to be higher at the shorter wavelengths
(i.e. < 550 nm), and they decrease more rapidly with increasing wavelength. The mean retrieved SSA values shown here are
within the range of the 550 nm ORACLES 2016 SSA values from multiple instruments presented in Pistone et al. (2019), but
are lower than most values from SAFARI 2000 (Haywood et al., 2003; Johnson et al., 2008; Russell et al., 2010). However,
the mean SSA is close to the 0.85 value reported by Leahy et al. (2007).  The lowest retrieved 550nm SSA value is only slightly
lower than that reported by Johnson et al., 2008 for the Dust and Biomass-burning Experiment (DABEX): 0.78 compared to
0.81.

Figure 4 compares our retrieved values of SSA to the *in situ* column average for a) 450 nm b) 530 nm and c) 660 nm for all
cases where such a comparison was possible. The *in situ* measurements are taken from a three-wavelength nephelometer (TSI
3563) and a three-wavelength particle soot absorption photometer (PSAP) (Radiance Research).  The combination of scattering
from the nephelometer and absorption from the PSAP provides SSA. SSA is calculated as the ratio of scattering from the
nephelometer to the sum of scattering (again from the nephelometer) and absorption (from the PSAP). In order to best compare
the retrieved values to the *in situ* values of SSA, the *in situ* measurements throughout the spiral profile are weighted by the
weighting function, obtained by the transmittance, and then averaged to obtain a column value of SSA. Further details of the
transmittance-weighted averaging can be found in Appendix C.

Although there are many factors that control aerosol SSA such as emission state, source location, distance from the source,
and age (Haywood et al., 2003; Eck et al., 2013; Konovalov et al., 2017; Dobracki et al., 2020 in prep), the values we find here
are well within the range of SSA values reported by other ORACLES instruments (Pistone et al., 2019). As seen in Figure 4,
the mean SSFR/4STAR retrieved SSA value tends to be slightly lower than the in-situ mean (shown by the blue curve on x-
and y- axis). However, there does not seem to be a distinct correlation or anti-correlation for these cases, especially considering
the uncertainties. This is consistent with the results shown in Pistone et al. (2019), which also showed no distinct correlation
between the SSA derived or measured by different instruments (top row in Figure 8).

It is important to note that the error bars shown in Figure 4 reflect different values between the instruments: the *in situ* error
bars represent the standard deviation of the entire column, whereas the SSFR-retrieved error bars represent the error estimate
of the retrieval. The *in situ* measurements provide a range of SSA, and the standard deviation illustrates the variability
throughout the aerosol layer. Conversely, the SSFR/4STAR retrieval provides only one value of SSA with the associated
retrieval uncertainty for the entire layer. The disadvantage to this is that we cannot detect any altitude dependence of SSA that
may be present.





## 3.2 DARE Parameterizations

The first (basic) parameterization $P_{DARE}$ uses only two input parameters: $AOD_{550}$ (mid-visible optical thickness) and $\alpha_{550}$

(scene or cloud albedo). The $L$ and $Q$ coefficients from Eq 12 are derived from the nine individual cases (described in section 3.3.1) where the corresponding fit coefficients for each of the cases are averaged to create the $P_{DARE}$ parameterization coefficients:

$$L_0 = \frac{1}{9}\sum_{i=1}^{9} l_{0,i}; L_1 = \frac{1}{9}\sum_{i=1}^{9} l_{1,i}; L_2 = \frac{1}{9}\sum_{i=1}^{9} l_{2,i},$$

$$Q_0 = \frac{1}{9}\sum_{i=1}^{9} q_{0,i}; Q_1 = \frac{1}{9}\sum_{i=1}^{9} q_{1,i}; Q_2 = \frac{1}{9}\sum_{i=1}^{9} q_{2,i}.$$

The coefficients $l_0$, $l_1$, $l_2$, $q_0$, $q_1$, $q_2$ are the linear ($l$) and quadratic ($q$) coefficients of second-order polynomial fits to radiative transfer calculations for the DARE dependence on $AOD_{550}$ of the *individual* cases as expressed in Eq 12 for the *average*, which simultaneously capture the dependence on $\alpha_{550}$ as follows:


$$l(\alpha_{550}) = l_0 + l_1 * \alpha_{550} + l_2 * \alpha_{550}^2, \tag{14}$$
$$q(\alpha_{550}) = q_0 + q_1 * \alpha_{550} + q_2 * \alpha_{550}^2, \tag{15}$$

The overall $P_{DARE}$ coefficients are tabulated for each solar zenith angle $SZA=\{0°, 5°, …, 80°\}$ (see Table 4a).

Figure 5a shows the dependence of $DARE = P(AOD_{550}, \alpha_{550})$ on the two input parameters for one specific SZA. DARE is shown in percent of top-of-atmosphere irradiance[1], $S_0 * cos\ (SZA)$, where $S_0 = 1361$ W/m². It is clearly non-linear with respect to both input parameters, illustrating the need for a quadratic representation. However, the RFE from which $P_{DARE}$ originates is still encapsulated in this parameterization as:

$$RFE = \frac{dP(AOD_{550}, \alpha_{550})}{dAOD_{550}}\Big|_{AOD_{550}=0} = L(\alpha_{550}), \tag{16}$$

which is the slope of the black line at the origin in Figure 5a. For a black surface, this reduces to $RFE = L_0$. In this sense, the full parameterization $P_{DARE}$ generalizes RFE.

Whereas the black lines in Figure 5a and 5b show the average ORACLES parameterization (i.e. $P_{DARE}$) from Table 4a, the

colored lines show the contributing 9 cases, sorted by 550 nm SSA. It is apparent that the SSA introduces considerable case-

---

[1] We will include supplementary material with the final version of the manuscript. This will include all necessary coefficients for the parameterization as well as the code necessary to reconstruct them. The specific solar constant can be defined by the user based on their input data.





to-case variability, both in terms of the critical albedo (Fig. 7) and in terms of the magnitude of the DARE, especially for large albedos (Fig. 6.)

To illustrate this point, Figure 6 shows the same as Figure 5b, but here as the difference between the DARE for individual cases and $P_{DARE}$, (which represents the case-average DARE) expressed as a percentage difference in incident TOA solar flux.

The $\pm\sigma$ range of variability (essentially the root mean square (RMS) difference, shown as dashed black lines in Figure 7) is calculated from the standard deviation of this difference across all nine cases enumerated by $c$:

$$\sigma = \sqrt{\frac{1}{8}\sum_{c=1}^{9}(DARE_c - \overline{DARE})^2}. \tag{17}$$

This serves as a metric for the case-to-case variability, which increases with the scene albedo and AOD. For example, the possible range in DARE for a mid-visible albedo of 0.6 and an AOD of 0.75 ($SZA=20°$) would be about 10±2% (or 136±27 Wm$^{-2}$). This is *without* accounting for the uncertainty in the input parameters AOD and scene albedo, which have to be propagated through the parameterization via $dP/dAOD$ and $dP/d\alpha$. The uncertainty of 27 Wm$^{-2}$ in brackets above can be interpreted as the uncertainty in DARE due to insufficient knowledge of SSA, which drives the case-to-case variability: in

Figures 5 and 6, the highest (lowest) SSA values correspond to the lowest (highest) DARE.

The extended parameterization $PX_{DARE}$ (equation 13) includes the SSA effect on DARE explicitly through an addition term not included in the $P_{DARE}$ parameterization (equation 12): $\Delta(AOD_{550}, \alpha_{550}, \Delta SSA_{550})$.

In order to quantify the effect of SSA by this term, it is convenient to start with the dependence of the critical albedo on SSA (Figure 7). To first approximation, this dependence can be represented by a linear fit. The critical albedo also weakly depends

on the AOD, and rather strongly on the SZA (not shown; for example, it can attain 0.6 at low Sun elevations). In contrast with the SSA, the asymmetry parameter does not drive the critical albedo in any discernible way, nor does it explain the deviation of the case-specific critical albedo from the fit line.

In analogy to the SSA dependence of the critical albedo, the case-specific deviations of DARE from the case-average DARE (Figure 6) can be represented as linear functions $\Delta(\alpha, SSA)$ (Figure 8a). Here, this is done by defining the DARE perturbation

$\Delta(SSA)$ at two specific albedos: (1) at the case-average critical albedo (i.e. the albedo where DARE changes sign in Figure 7), and (2) an albedo of 1 (maximum albedo):

$$\Delta_{crit} = \Delta\left(AOD_{550}, \alpha_{550}^{crit}, \Delta SSA_{550}\right) = C(AOD_{550}) * \Delta SSA \tag{18}$$

$$\Delta_{max} = \Delta\left(AOD_{550}, \alpha_{550}^{max}, \Delta SSA_{550}\right) = D(AOD_{550}) * \Delta SSA \tag{19}$$


where C and D are the slopes of the fit lines of $\Delta(\alpha, SSA)$ and $\Delta SSA$ is the difference between the case-specific SSA and the case-average SSA ($\overline{SSA}$, 0.83). The colored dots in Figure 8a show $\Delta_{crit}$ and $\Delta_{max}$, while Figure 8b shows how the coefficients C and D depend on the AOD. This dependency can be represented as:





$C(AOD) = C_1 * AOD + C_2 * AOD^2$         (20)

$D(AOD) = D_1 * AOD + D_2 * AOD^2,$         (21)

where $C_1$, $C_2$, $D_1$, and $D_2$ are tabulated in Table 4b for all solar zenith angles. Inserting Eqs. 20 into 18 and 21 into 19, the perturbations $\Delta_{crit}$ and $\Delta_{max}$ become:


$\Delta_{crit}(AOD_{550}, SSA_{550}) = (C_1 * AOD + C_2 * AOD^2) * (SSA - \overline{SSA})$     (22)

$\Delta_{max}(AOD_{550}, SSA_{550}) = (D_1 * AOD + D_2 * AOD^2) * (SSA - \overline{SSA})$     (23)

The perturbation at *any* albedo between the critical albedo and 1 is simply calculated as:


$\Delta(\alpha) = \frac{\alpha - \alpha_{crit}}{1 - \alpha_{crit}} * \Delta_{max} + \frac{1 - \alpha}{1 - \alpha_{crit}} * \Delta_{crit}$ ,     (24)

while $\Delta(\alpha) = \Delta_{crit}$ for $\alpha < \alpha_{crit}$.

Equations 21, 22, 23, and 24 are used collectively to determine the additional term for the $PX_{DARE}$ parameterization (Eq. 13).

If SSA is known in addition to AOD and scene albedo, then $PX_{DARE}$ captures DARE to greater fidelity than does $P_{DARE}$. This is shown by the case-to-case variability in Figure 9, expressed as the difference between the DARE for the individual cases $PX(AOD_{550}, \alpha_{550}, \Delta SSA_{550})$ in analogy to Figure 6. The $\pm\sigma$ range of variability in Figure 9 is much smaller than that in Figure 6, showing that the uncertainty in $PX_{DARE}$ ($\pm 0.5\%$ at an albedo of 0.3 of the incident irradiance at TOA) is significantly below the unresolved variability in $P_{DARE}$ due to an unknown SSA ($\pm 1.2\%$ at an albedo of 0.3, up to 2% at an SZA of 20°).

Beyond the case-to-case variability, Figure 10 confirms that including the SSA information in $PX_{DARE}$ does in fact reproduce DARE well for each individual case, as illustrated by the agreement between the solid ($PX_{DARE}$) and the individual case RTM-calculated DARE. The residuals between the direct RTM DARE output and DARE estimated using $P_{DARE}$ and $PX_{DARE}$ (shown as contours in Figure 11a and Figure 11b) provide an estimate of the overall uncertainties inherent within the parameterizations.

As Figure 11a shows, the residuals of $PX_{DARE}$ are significantly smaller than those of $P_{DARE}$. Both $P_{DARE}$ and $PX_{DARE}$ have small uncertainty contributions from a number of factors (e.g., measurement uncertainty of SSFR, RTM uncertainty, conversion and extrapolation from spectrally resolved retrievals to broadband values, the uncertainty of the quadratic fit leading to the $L$ and $Q$ coefficients, and the uncertainty in the fits leading to the $C$ and $D$ coefficients), but $P_{DARE}$ also encompasses the variability due to SSA which leads to a much larger uncertainty in $P_{DARE}$ than $PX_{DARE}$.



## 4. Summary and Interpretation


In this paper, we systematically linked aircraft observations of spectral fluxes to aerosol optical thickness and other parameters, using 9 cases from the 2016 and 2017 ORACLES campaigns. This observationally-driven link is expressed by two parameterizations of the shortwave broadband DARE, (1) in terms of the mid-visible AOD and scene albedo ($P_{DARE}$), and (2) in terms of the mid-visible AOD, scene albedo, and aerosol SSA ($PX_{DARE}$). These parameterizations can be used to translate

from AOD and scene albedo (optionally also from SSA) to DARE directly, bypassing radiative transfer calculations that are usually required to arrive at DARE from observations. This is advantageous when satellite retrievals provide only limited information such as AOD and scene albedo (by way of cloud fraction and optical thickness), but not aerosol microphysics, hygroscopic growth, or optical properties. However, this parameterization only captures the natural variability of the study region as sampled. It therefore does not necessarily represent the entire southeast Atlantic, let alone during times beyond the

ORACLES campaigns. Despite this caveat, one could interpret the parameterization as the start of a DARE climatology built on two (or three) driver variables. Additional observations extending the statistics to other regions and time periods could easily be added to this framework. For example, the 2018 ORACLES data will be incorporated in a separate paper.

We find that the two parameterizations reproduce the case-specific DARE to different degrees. The majority of the case-to-case variability within the ORACLES DARE dataset is attributable to the dependence on AOD and scene albedo. Using just

these two variables to span the first parameterization, $P_{DARE}$, the RMS bias of the case-specific DARE with respect to the parameterized baseline is 1-2% of the incident radiation for an SZA of 20° and an AOD of 0.75 (Figure 6), with a DARE value of 10% of the incident radiation for a scene albedo of 0.6 (Figure 5b). Translated into flux units, the DARE for this constellation of scene parameters is 136±27 W m$^{-2}$, where the range of uncertainty stems from the unexplained case-to-case variability as obtained from the RMS bias. In other words, this parameterization leads to 20% DARE uncertainty due to the variability of

the system caused by factors *other* than AOD and scene albedo. If satellites only provided AOD and scene albedo, this would be the uncertainty of the derived DARE (leaving the retrieval uncertainties of AOD and albedo aside for the moment). In reality, the variability is likely even larger than captured with our limited samples, so this estimate is a lower bound on the DARE variability.

Fortunately, our research showed that we can actually explain more of the case-to-case variability by introducing the mid-

visible SSA as third parameter in an extended parameterization $PX_{DARE}$. This reduces the variability by a factor of 4 by explicitly resolving the case-to-case variability via SSA: a DARE value of 136±6.8 Wm$^{-2}$ corresponds to an SSA of 0.83 (campaign average at 550 nm), whereas 0.81 (typical low SSA value encountered during ORACLES) yields DARE of 177±10.6 Wm$^{-2}$. The remaining uncertainty (about 5%) is due to variability drivers beyond AOD, scene albedo and SSA, such as variable aerosol microphysics or hygroscopicity. It also encompasses the measurement uncertainty of SSFR and 4STAR.

Interestingly, the mid-visible asymmetry parameter (also retrieved for most cases) is not a significant driver of the case-to-case variability. However, the retrieved spectra of SSA and asymmetry parameter can be useful for future satellite retrievals of cloud and aerosol optical thickness in the study region. Since these retrievals are directly tied to the radiative fluxes, they



work without assumptions about the scattering phase function, size distribution, or aerosol type, nor do they require smoothness constraints. However, an optical closure study that involves in-situ measurements of aerosol microphysics and optical
properties in conjunction with Mie calculations is required before our results can be of practical use, especially at wavelengths beyond the visible range where our retrieval uncertainties grow large. Our asymmetry parameter spectra fall off faster with wavelength than usually assumed based on land-based observations, which may be an indication that there is less coarse mode in the ORACLES measurements, which are almost exclusively over ocean.

We cannot judge whether our approach will be useful for predictive models, which usually follow the "bottom-up" paradigm,
i.e., they arrive at DARE starting from detailed aerosol and cloud properties via radiative transfer calculations. At the very least, the SSA and asymmetry parameter retrievals coming out of our and other ORACLES studies will constrain the aerosol optical properties in a range of models. However, we also anticipate that our parameterized, observationally-based DARE could serve as a simple, built-in closure for the calculated DARE, adding a "top-down" model constraint, or even prove useful for model tuning.

Our paper stops short of providing an "all-ORACLES" DARE estimate. Such studies are ongoing and will be published separately. Satellites will be required to derive a regionally representative estimate, and to integrate instantaneous DARE to diurnally- and seasonally-averaged values. A grid-box specific model-observation inter-comparison is under way, and we expect that it will entail detailed radiative and optical closure efforts. While we limited this paper to the above-layer (TOA) DARE, the radiative effect of aerosols on the layer itself (i.e., the heating rate) is also an important deliverable from ORACLES,
which will be presented in a separate follow-up paper.

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

**Spiral Data Conditioning**

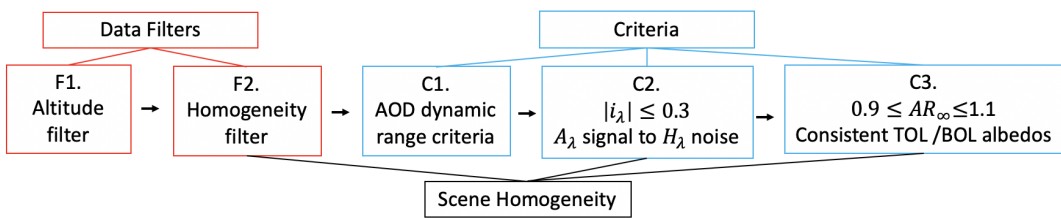

**Figure 1: Data conditioning flow chart. First, the data is filtered vertically (i.e. data is removed) to F1) isolate the aerosol layer only and F2) isolate either cloudy or clear sky data such that the profile represents a homogeneous sky type. Once filtered, the data must pass 3 distinct criteria to ensure that C1) the full aerosol layer is captured C2) the effect of aerosol absorption on radiative fluxes is much greater than that due to horizontal variability present and C3) the top-of-layer (TOL) and bottom of layer (BOL) albedos are mutually consistent.**


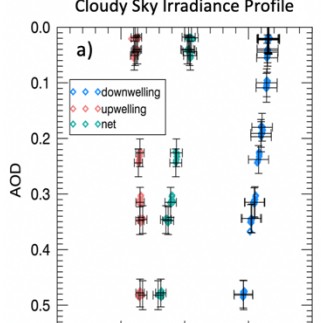 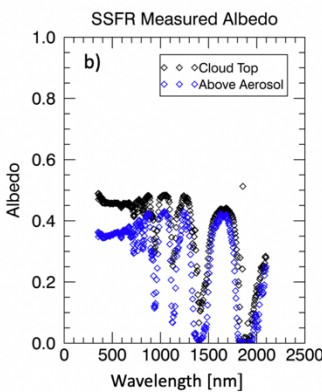 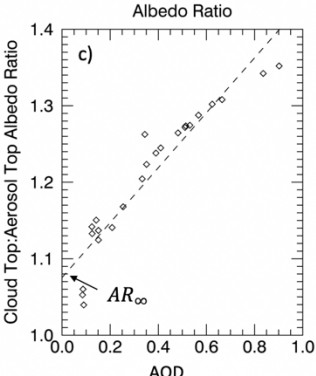

**Figure 2: a) Above cloudy sky upwelling, downwelling, and net irradiance profiles shown versus the measured 532 nm AOD by 4STAR with associated measurement error bars for one example case. The AOD refers to the air above the aircraft and generally decreases with increasing aircraft altitude, hence the inverted y-axis. b) SSFR measured albedo spectrum at the bottom of the spiral**
**(cloud top) and at the top of the spiral (above the aerosol layer.) c) The ratio between the BOL and TOL albedo spectra shown against the BOL AOD spectrum. The intercept of the fit line is criteria 3 ($AR_\infty$); if the intercept deviates largely from 1.0, the case cannot be used for an aerosol retrieval.**



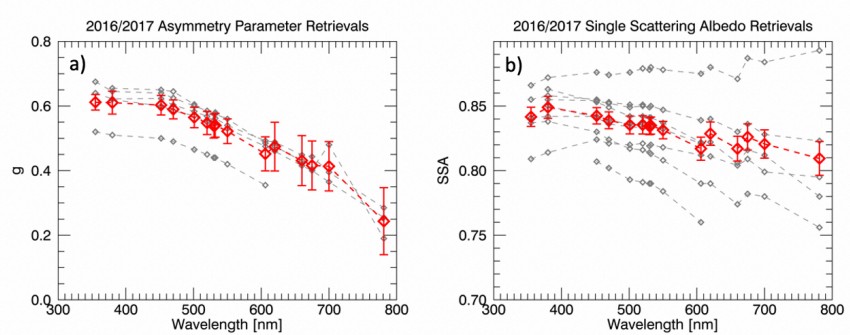

**Figure 3: Retrieved a) asymmetry parameter and b) SSA spectra for 2016 and 2017 successful retrievals. The red spectrum indicates the mean retrieved values with associated error bars; the grey spectra are the individual retrievals.**

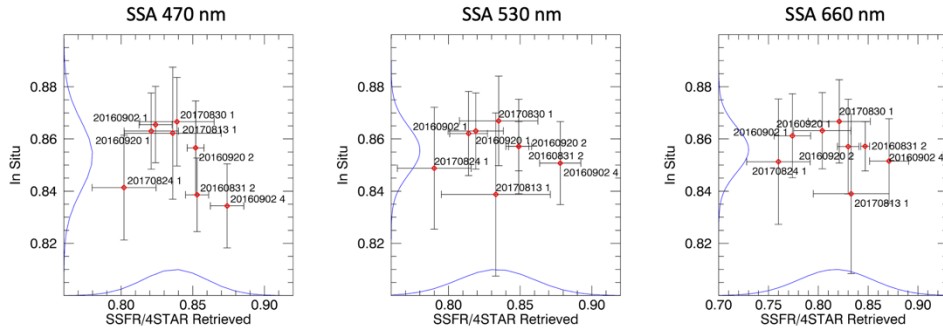

**Figure 4: *In situ* vs. retrieved SSA values for a) 470 b) 530 and c) 660 nm. *In situ* values show transmittance-weighted SSA representative of the whole column, with error bars representing the standard deviation of all measured values throughout the spiral profile. *In situ* data is not available for the 20170812 case and is therefore not shown. The uncertainties for retrieved SSA for all wavelengths are provided in Appendix E.**





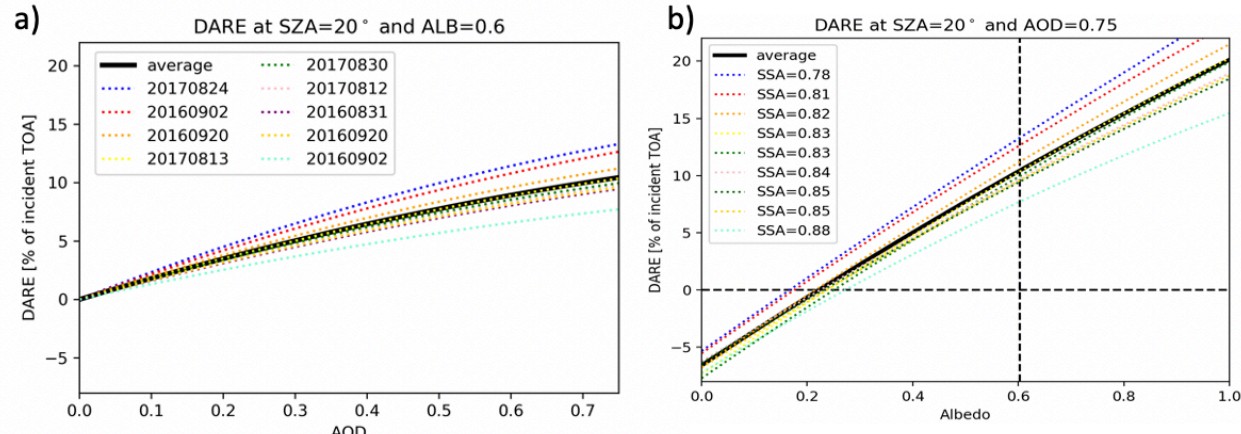

**Figure 5: (a) DARE as a function of AOD for fixed surface albedo (0.6) and SZA (20°), shown for the individual 9 cases from this**
**study (colors) and the average (black). The average is the basic parameterization result, P$_{DARE}$. (b) DARE as a function of surface albedo for a fixed AOD (0.75). The individual cases are labelled by their SSA at 550 nm (from more to less absorbing). The albedo at which the DARE changes is the critical albedo (horizontal dashed line). The vertical line marks an albedo of 0.6 for much of the ensuing discussion, which uses an AOD of 0.75, an albedo of 0.6, and a SZA of 20°.**

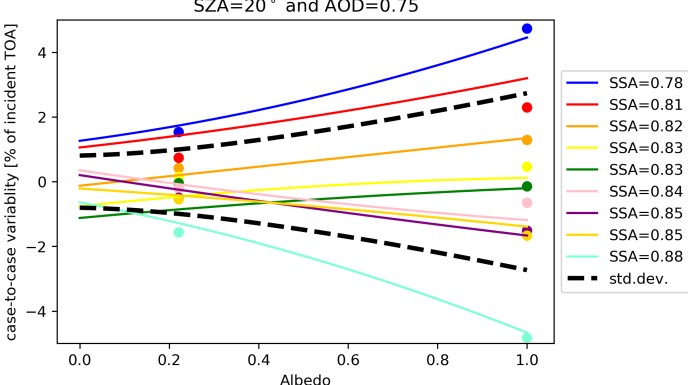

**Figure 6: The difference between P$_{DARE}$ and DARE for the individual cases at a fixed AOD (0.75) and SZA (20°). The range of variability is represented by the standard deviation (black dashed curves).**




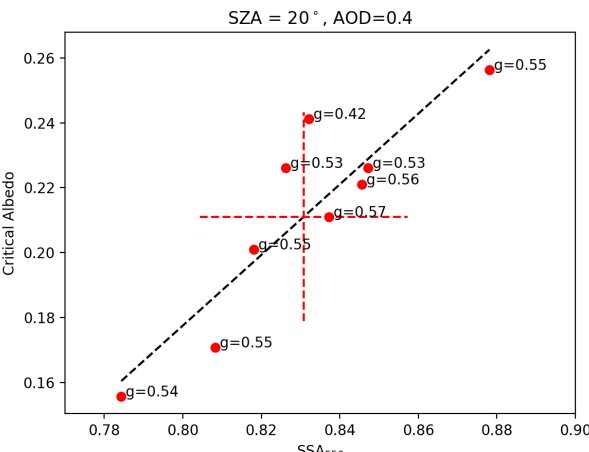

**Figure 7. Critical albedo as a function of mid-visible SSA. The red dashed cross shows the case-average $\alpha_{crit}$.**

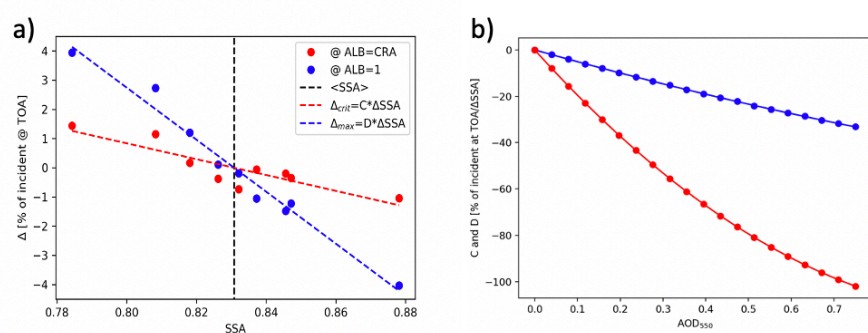

**Figure 8: (a) DARE perturbations as a function of SSA at the case-average critical albedo (red) and at albedo=1 (blue) for SZA=20°. The vertical black dashed line indicates the case-average SSA. (b) the dependence of the parameters C (red curve; determined at the critical albedo (Eq. 19) and D (blue curve; determined at albedo=1 (Eq. 20) coefficients on mid-visible AOD.**





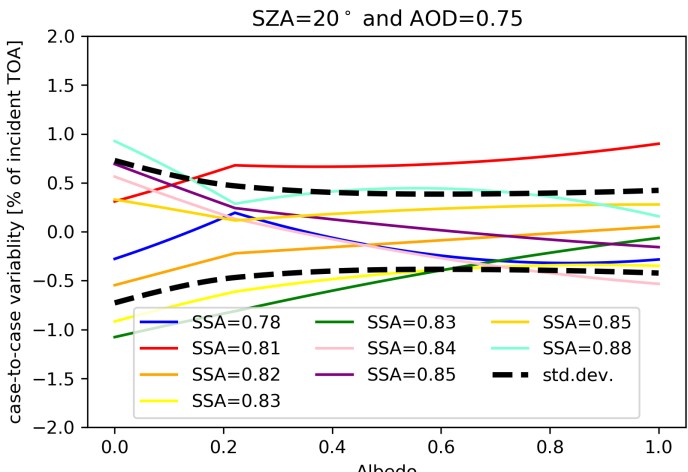

Figure 9: The difference between $PX_{DARE}$ and $P_{DARE}$ for 9 case SSAs at fixed AOD (0.75) and SZA (20°).

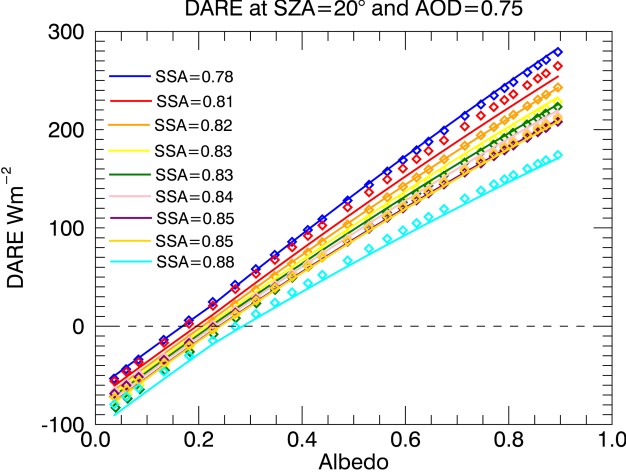

Figure 10: DARE as predicted by $PX_{DARE}$ for the nine cases (solid lines) and DARE as calculated by the RTM (dotted colored lines).



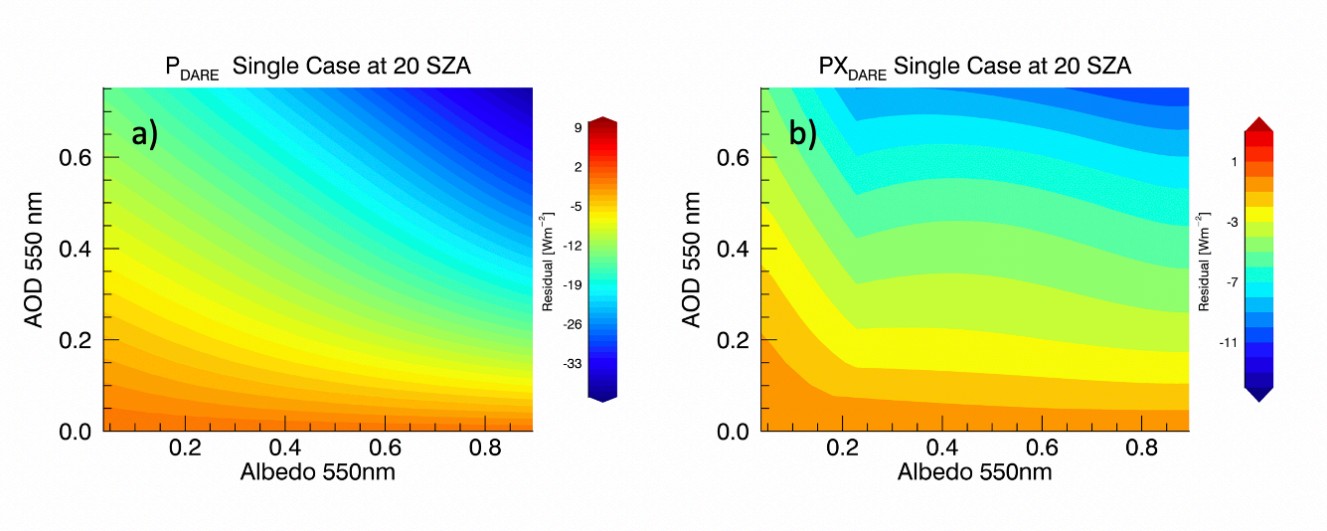

**Figure 11: Residual plot of directly calculated DARE (RTM output) and predicted BB DARE values using (a)P$_{DARE}$and (b)PX$_{DARE}$for a single case at a fixed SZA (20°). Residual plots for each case can be found in Appendix D. For both figures, the residuals encompass the difference between the RTM and the P$_{DARE}$ and PX$_{DARE}$ parameterizations.**

| Date | C2: longest retrievable wavelength [nm] for which $|i| < 0.3$. | C3:$AR_\infty$ | Status: $SSA_\lambda$ /$g_\lambda$ |
|---|---|---|---|
| 20160831 #1* | Fail | | |
| 20160831 #2* | 550 nm | 1.04 | yes/yes |
| 20160902 #1 | >781nm | 1.01 | yes/yes |
| 20160902 #4 | >781 nm | 0.98 | yes/no |
| 20160910 #1 | Fail | | |
| 20160920 #1 | 781 | 1.02 | yes/no |
| 20160920 #2 | 781 | 1.07 | yes/yes |





| 20160924 #1 | 1627 | Fail | |
|---|---|---|---|
| 20160924 #3 | Fail | | |
| 20160927 #1 | Fail | | |
| 20170809 #1 | Fail | | |
| 20170809 #2 | >781 | Fail** | |
| 20170812 #1 | Fail | | |
| 20170812 #3 | 781 | 1.02 | yes/yes |
| 20170813 #1* | 520 | 1.02 | yes/no |
| 20170815 #1 | 675 | Fail | |
| 20170824 #1 | 606 | 1.05 | yes/no |
| 20170826 #1 | 355 | Fail | |
| 20170826 #3 | Fail | | |
| 20170828 #1 | 1559 | Fail** | |
| 20170830 #1 | 606 | 1.07 | yes/yes |
| 20170831 #1 | Fail | | |

**Table 1: Retrieval Quality Metrics. Spirals are listed by date and the number in which they were performed on a particular flight.**
**Spiral cases that did not have data spanning the entire aerosol layer are excluded from the chart (i.e. did not pass criteria #1.) The**
**second column lists the longest wavelength for which $i_\lambda$ remains below 0.3; the aerosol retrieval is only valid up to this wavelength.**
**If $i_\lambda$ at all wavelengths is greater than 0.3, the case fails completely. The third column lists the $AR_\infty$ value. The intercept must fall**
**between 0.9 and 1.1 to pass this metric. The right-most column provides the status for the retrieval of $SSA_\lambda$ and $g_\lambda$. Cases that are**
**analyzed using the mean fit rather than the updated linear fit (update 2 from Cochrane et al., 2019) are indicated by *. Cases that**
**pass a metric but have a bad spectral shape in the albedo ratio (indicating failure) are indicated by **.**





| Date | UTC range | Latitude (mean) | Longitude (mean) | Cloud Albedo [500 nm] | Solar Zenith Angle | AOD [500 nm] | Column water vapor [g/cm$^2$] | Column ozone [DU] |
|---|---|---|---|---|---|---|---|---|
| 20160831 #2 | 13:12-13:33 | -17.2 | 7.04 | 0.69 | 37.2 | 0.6 | 1.04 | 289.7 |
| 20160902 #1 | 10:12-10:30 | -15.94 | 8.96 | 0.6 | 28.5 | 0.42 | 1.1 | 342.3 |
| 20160902 #4 | 12:09-12:27 | -15.02 | 8.53 | 0.65 | 26.2 | 0.46 | 1.31 | 341.7 |
| 20160920 #1 | 9:09-9:21 | -16.73 | 10.55 | 0.73 | 33.8 | 0.47 | 0.87 | 410.6 |
| 20160920 #2 | 11:52-12.15 | -16.68 | 8.9 | 0.45 | 21.2 | 0.57 | 1.15 | 441.9 |
| 20170812 #3 | 14:30-14:57 | -2.9 | 5.04 | 0.57 | 46.7 | 0.32 | 1.37 | 243.8 |
| 20170813 #1 | 10:00-10:30 | -8.97 | 4.95 | 0.7 | 33.6 | 0.21 | 0.41 | 268.8 |
| 20170824 #1 | 11:00-11:30 | -14.9 | 5.1 | 0.54 | 26.4 | 0.27 | 0.77 | 326.2 |
| 20170830 #1 | 12:20-13:00 | -8.05 | 4.91 | 0.49 | 23.2 | 1.36 | 1.6 | 290.9 |

**Table 2. Spiral case details for successful aerosol retrievals. The albedo, SZA, AOD, column water vapor and column ozone are**
**used within the radiative transfer model to retrieve aerosol properties and calculate DARE. The AOD, water vapor, and ozone are**
**all reported above cloud.**





| Wavelength [nm] | 355 | 380 | 452 | 470 | 501 | 520 | 530 | 532 | 550 | 606 | 620 | 660 | 675 | 700 | 781 |
|---|---|---|---|---|---|---|---|---|---|---|---|---|---|---|---|
| $n_{SSA}/n_g$ | 5/3 | 8/5 | 9/5 | 9/5 | 9/5 | 9/5 | 8/5 | 8/5 | 8/5 | 7/4 | 5/3 | 5/3 | 5/3 | 5/3 | 5/3 |
| SSA | 0.84 | 0.85 | 0.84 | 0.84 | 0.84 | 0.84 | 0.83 | 0.83 | 0.83 | 0.82 | 0.83 | 0.82 | 0.83 | 0.82 | 0.81 |
| $\sigma_{SSA}$ | 0.02 | 0.02 | 0.02 | 0.02 | 0.02 | 0.02 | 0.03 | 0.03 | 0.03 | 0.04 | 0.03 | 0.04 | 0.04 | 0.04 | 0.05 |
| g | 0.61 | 0.61 | 0.6 | 0.59 | 0.56 | 0.55 | 0.54 | 0.54 | 0.52 | 0.45 | 0.47 | 0.43 | 0.42 | 0.41 | 0.24 |
| $\sigma_g$ | 0.08 | 0.06 | 0.06 | 0.06 | 0.06 | 0.06 | 0.06 | 0.06 | 0.06 | 0.06 | 0.01 | 0.02 | 0.02 | 0.06 | 0.05 |

**Table 3. Mean retrieved SSA (row 3) and *g* (row 5) spectra along with their associated standard deviations ($\sigma$) (row 4, row 6, respectively). The second row provides the number of valid retrievals for that wavelength. As described in Cochrane et al., 2019, individual wavelengths can fail within the retrieval resulting in fewer valid retrievals than valid cases (e.g. 355 nm SSA has 5 valid retrievals despite having 9 valid cases).**

| SZA | L0 | L1 | L2 | Q0 | Q1 | Q2 |
|---|---|---|---|---|---|---|
| 0° | -135.2±17.1 | 751.1±57.3 | -168.1±26.1 | 31.6±6.1 | -269.8±31.9 | 126.7±17.2 |
| 10° | -136.1±17.1 | 743.3±56.6 | -164.8±25.7 | 32.36±6.1 | -268.1±31.6 | 124.6±16.9 |
| 20° | -138.5±16.9 | 720.2±54.6 | -154.9±24.4 | 34.6±6.2 | -263.3±30.9 | 118.4±16.2 |
| 30° | -142.5±16.6 | 682.4±51.2 | -138.8±22.4 | 38.7±6.4 | -255.7±29.5 | 108.0±14.9 |
| 40° | -147.9±16.1 | 630.5±46.5 | -117.0±19.6 | 45.1±6.5 | -246.1±27.6 | 93.5±13.2 |
| 50° | -153.9±15.2 | 564.5±40.5 | -90.4±16.2 | 54.7±6.5 | -235.1±24.9 | 74.7±11.1 |
| 60° | -158.0±13.7 | 482.5±33.3 | -60.7±12.1 | 67.5±6.2 | -221.6±21.2 | 51.9±8.6 |
| 70° | -152.6±11.6 | 378.5±25.5 | -32.7±7.6 | 80.3±6.7 | -200.5±16.8 | 27.1±5.7 |
| 80° | -116.0±8.6 | 239.4±18.2 | -24.1±3.4 | 77.4±5.6 | -156.8±13.1 | 15.5±2.5 |

**Table 4a. P$_{DARE}$ parameterization coefficients for differing SZAs. The collection of the coefficients represent the mean of all cases and the uncertainty values represent the standard deviation; the units on the *L* coefficients are W/m²/unit optical depth; the units on the *Q* coefficients are W/m²/(unit optical depth)²**





| SZA | C1 | C2 | D1 | D2 |
|-----|------|-------|---------|--------|
| 0° | -652.0 | 113.5 | -2741.3 | 1210.9 |
| 10° | -657.0 | 120.9 | -2712.1 | 1201.6 |
| 20° | -665.3 | 129.0 | -2625.2 | 1174.6 |
| 30° | -682.0 | 151.2 | -2482.3 | 1131.2 |
| 40 | -703.2 | 185.6 | -2284.9 | 1072.8 |
| 50° | -725.6 | 235.7 | -2032.9 | 999.9 |
| 60° | -733.1 | 295.9 | -1721.7 | 909.4 |
| 70° | -692.5 | 353.7 | -1335.1 | 787.5 |
| 80° | -524.3 | 353.2 | -817.6 | 577.3 |

**Table 4b PX$_{DARE}$ additional coefficients for differing SZAs. These coefficients are used in Equations 22 and 23 (inserted into Equation 24) and act as extension to P in order to resolve the case-to-case variability resolvable through SSA. The units on the *C1* and *D1* coefficients are W/m$^2$/unit optical depth; the units on the *C2* and *D2* coefficients are W/m$^2$/(unit optical depth)$^2$**

**Appendix A. Extension from spectral to broadband**

Making the transition from the spectral to broadband is one of the main hurdles for both the parameterizations presented in this paper and for broadband DARE studies in general. Broadband DARE calculations require accurate aerosol and cloud information for all wavelengths, and it can be difficult to accurately determine the correct spectral dependence of these properties. The cloud albedo is particularly challenging since the spectral dependence depends on the SZA.

In our work, the aerosol optical properties of SSA and $g$ can be retrieved for wavelengths up to 781 nm, and AOD values from
4STAR can be retrieved for up to 1650 nm. Cloud albedo is measured for the entire SSFR wavelength range, but only for a single SZA value (the mean SZA throughout the spiral time period). We therefore must a) interpolate between wavelengths and b) extend each optical property to longer wavelengths to the best of our knowledge and compute the cloud albedo for a range of SZAs.



### A.1 SSA

To extend the retrieved SSA values to the remaining reported 4STAR wavelengths, we rely on the AAOD, defined as:

$$AAOD_\lambda = AOD_\lambda * (1 - SSA_\lambda). \hspace{2cm} (1A)$$

First, we calculate a fit line of the AAOD for wavelengths where we have valid SSFR SSA retrievals. We then extend that fit to obtain the AAOD for the remaining 4STAR wavelengths. We then re-arrange Equation 1A to determine SSA for those wavelengths where we do not have SSFR SSA retrievals. Finally, we set the SSA at wavelengths longer than 1650 nm to the

mean of the longest 4STAR wavelengths, 1600 and 1650 nm. A1a illustrates the extension of SSA.

### A.2 Asymmetry Parameter

Using the SSFR retrieved *g* values, we calculate a polynomial fit for the available wavelengths. We then extend the fit to longer wavelengths. Once the fit reaches 0, the remaining wavelengths are set to 0. While it would have been possible to instead use the fine mode Mie calculations (Figure B1), we chose to utilize the retrievals and approximate the fine mode, jumping to zero

lacking other information. An optical closure study, though beyond the scope of this paper, is necessary. Figure A1b illustrates the extension of g.

### A.3 Developing the Parameterization Grid

In order to calculate the parameterization, we grid the AOD and albedo spectra, preserving the specific spectral shapes.

### A.3.1 AOD

We take the measured AOD spectrum at the BOL and multiply that spectrum by a factor to create a grid such that the values at 550 nm range from 0 to 0.75. In this way, each case has a normalized AOD grid at 550 nm while maintaining the specific spectral shape of the measured spectrum. We then extrapolate the AOD spectra to the remaining wavelengths. Figure A1c illustrates the extension and gridding of AOD.

### A.3.2 Albedo

Obtaining the cloud albedo requires using the RTM to maintain accurate representation of the spectral shape. First, we retrieve the cloud properties of effective radius (Reff), and cloud optical thickness (COT) from the measured albedo using the RTM, with retrieval wavelengths of 1200 nm and 1630 nm. We then grid COT from 0 to 100 while keeping Reff constant at the retrieved value. We run the RTM to calculate a spectral albedo grid for all new pairs of Reff and COT for the range of SZAs. Figure A1d illustrates the albedo grid for a single SZA.

While we extend the aerosol and cloud properties as accurately as possible, it is most crucial that the shortest wavelengths are accurate. At the longer wavelengths, the AOD becomes increasingly small, and the optical property accuracy is therefore less





critical. This works in our favor since the SSFR retrieval is valid for this wavelength range where the AOD and absorption are large.

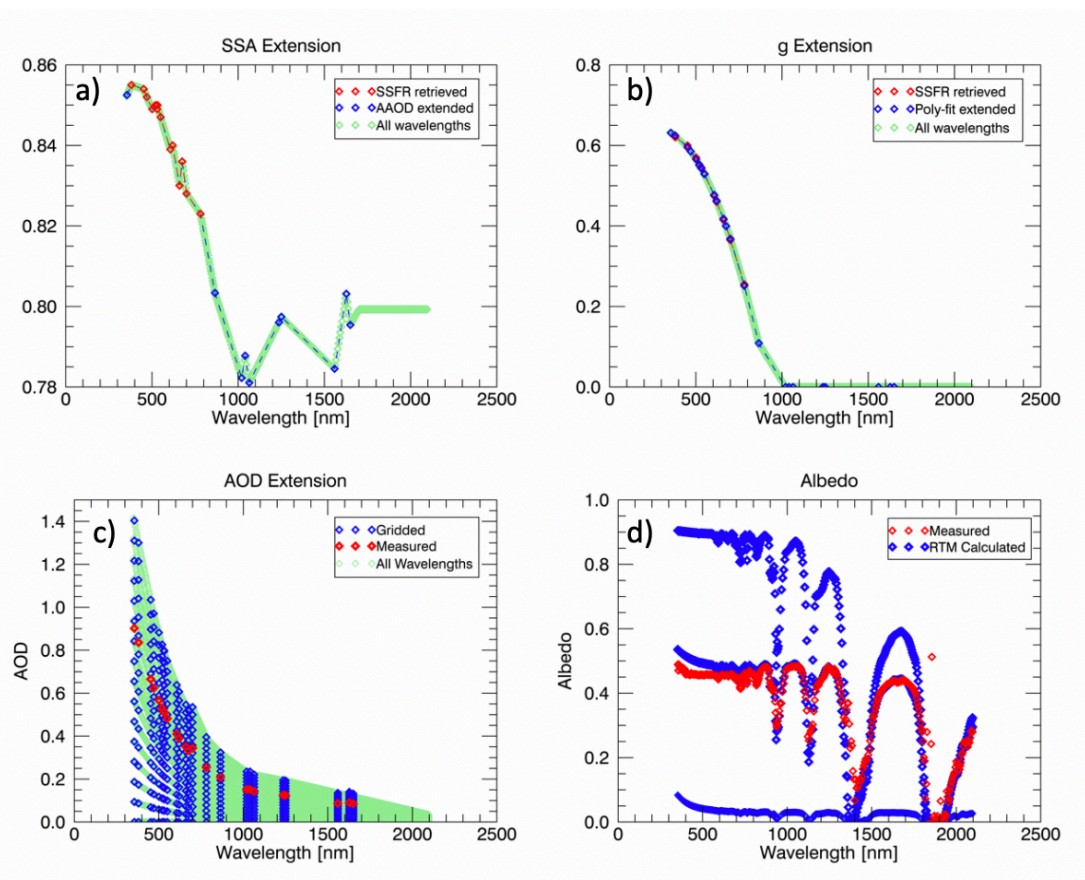

**Figure A1.** One example case of the extension of aerosol properties to longer wavelengths for a) SSA b) *g* and c) AOD. d) Shows the SSFR measured vs. RT-calculated albedo spectra along with the RT-calculated spectra for 0 COT and 100 COT.

### Appendix B. Irradiance Retrieval

The SSFR spectral irradiance aerosol retrieval is fundamentally different than most other aerosol retrievals, which are rooted in knowledge of the aerosol size distribution along with both the imaginary and real parts of the index of refraction. These methods must utilize Mie calculations to get to the aerosol optical properties of SSA and g. As described in Pistone et al., (2019), ORACLES instrumentation such as 4STAR, the Research Scanning Polarimeter (RSP), and the Airborne Multi-angle SpectroPolarimeter Imager (AirMSPI) utilize this technique to obtain aerosol properties. The SSFR retrieval, on the other





hand, circumvents the need for Mie calculations and knowledge of the size distribution or index of refraction by relying on the measured aerosol absorption itself.

However, simple Mie calculations (Figure B1) verify that a quickly decreasing asymmetry parameter is possible, and it will even decrease to 0 if no coarse mode is present. However, that is unlikely. It is more likely that the asymmetry parameter eventually goes back up again for long wavelengths - a result of even small coarse mode concentrations.

Beyond the ORACLES-specific instrumentation, AERONET stations across the globe utilize sunphotometers with the same underlying retrieval algorithms as used with 4STAR sky radiances to provide aerosol optical properties. In figure B2a and B2b, we show the mean SSFR SSA and *g* retrieval spectra compared to the nearest AERONET sites for 2016 and 2017: São Tomé, Ascension and Namibia.

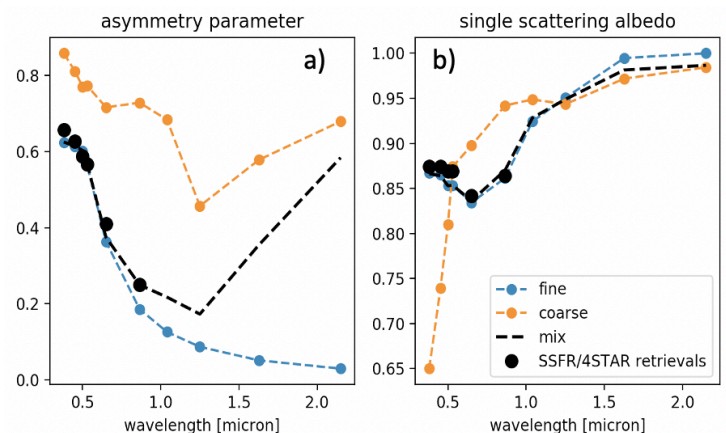

**Figure B1.** Mie calculations of (a) *g* and b) SSA compared to SSFR/4STAR retrieved values. The black dots show the asymmetry parameter spectrum (left) and SSA spectrum (right) as retrieved from SSFR/4STAR; the blue dot-dash line shows a fine-mode aerosol (r=0.13 micron) with real index of refraction of 1.6, and imaginary index of refraction ranging from 0.05 (380nm) to 0 (2micron); the orange dot-dash line shows a coarse-mode aerosol (r=1.3 micron) with real index of refraction of 1.6, and imaginary index of refraction ranging from 0.015 (380nm) to 0.003 (600nm) [Wagner et al., 2012]; The black line shows a mix of coarse/fine
aerosol (0.02:2 optical thickness ratio).

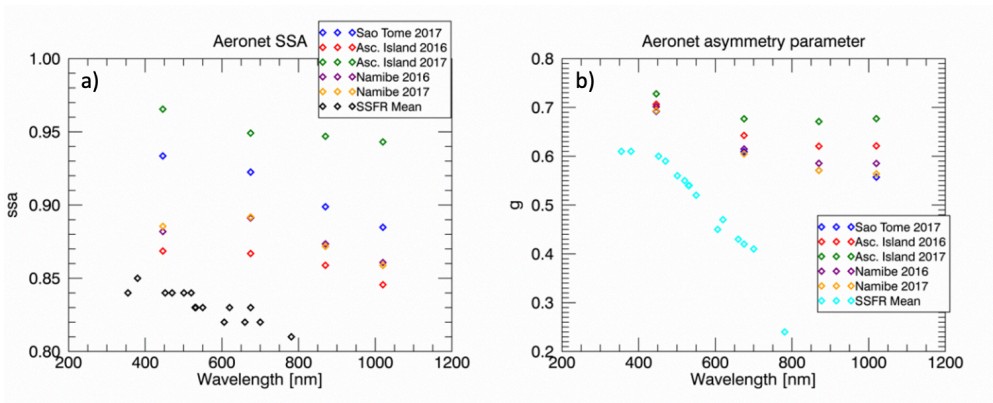

**Figure B2.** Retrieved values of a) SSA and b) *g* compared AERONET measured values at nearby land sites.





## Appendix C. *In Situ* Transmittance Weighting

*In situ* SSA measurements and SSFR SSA retrievals cannot be compared directly since *in situ* SSA measurements are made
continuously throughout the column (spiral), across variations in aerosol concentrations, whereas the SSFR SSA values
represent a single value representative of the entire column. In order to best compare the *in situ* and retrieved SSA values, we
calculate a weighted *in situ* SSA average, using a weighting function based on the transmittance through the aerosol layer.
In past studies, (e.g. Cochrane et al., 2019; Pistone et al., 2019) the *in situ* SSA measurements were averaged with each SSA
value weighted by its corresponding measured extinction which better represents the column SSA than a simple average.
However, it is the transmittance rather than the extinction which describes the aerosols' impact on the radiation throughout the
layer. Since the SSFR SSA retrieval is based on the change in radiation through the aerosol layer, it is most consistent to weigh
the *in situ* measurements on transmittance rather than extinction.

For each spiral profile, we take the extinction profile as measured by the *in situ* instruments to calculate the weighting function
as follows:

$$W(z) = \frac{\beta_e(z)}{\mu} e^{-\frac{\tau(z)}{\mu}} = \frac{\beta_e(z)}{\mu} t(z).$$

where $\beta_e(z)$ is the extinction, $t(z)$ is the transmittance, and $\mu = \frac{1}{cos(SZA)}$.

Figure C1 shows the *in situ* measured SSA profile for one profile case at a) 470 nm b) 530 nm and c) 660 nm. The red dashed
line shows the SSFR/4STAR retrieved value; the black dashed line shows the transmittance-weighted *in situ* SSA value; the
gray dashed line shows the extinction weighted *in situ* SSA value.

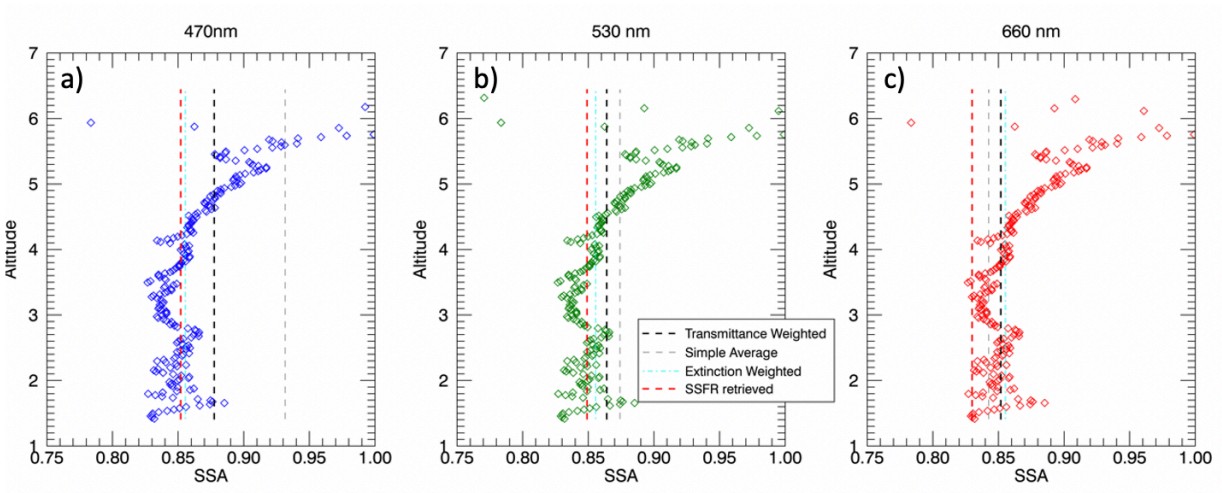


**Figure C1. An example of one spiral case with the different *in situ* averages along with the SSFR retrieved SSA for a) 450 nm b) 530
nm and c) 660 nm. The colored points show the *in situ* data as measured throughout the profile.**



**Appendix D: Residual Figures.**


Figures D1 and D2 show the residual values between directly calculated DARE (by the RTM) and DARE calculated using D1) $P_{DARE}$ and D2) $PX_{DARE}$ for each case. The residuals are significantly higher when using $P_{DARE}$ vs. $PX_{DARE}$, illustrating that including the additional constraint of SSA (i.e. $PX_{DARE}$) greatly improves the parameterization performance.

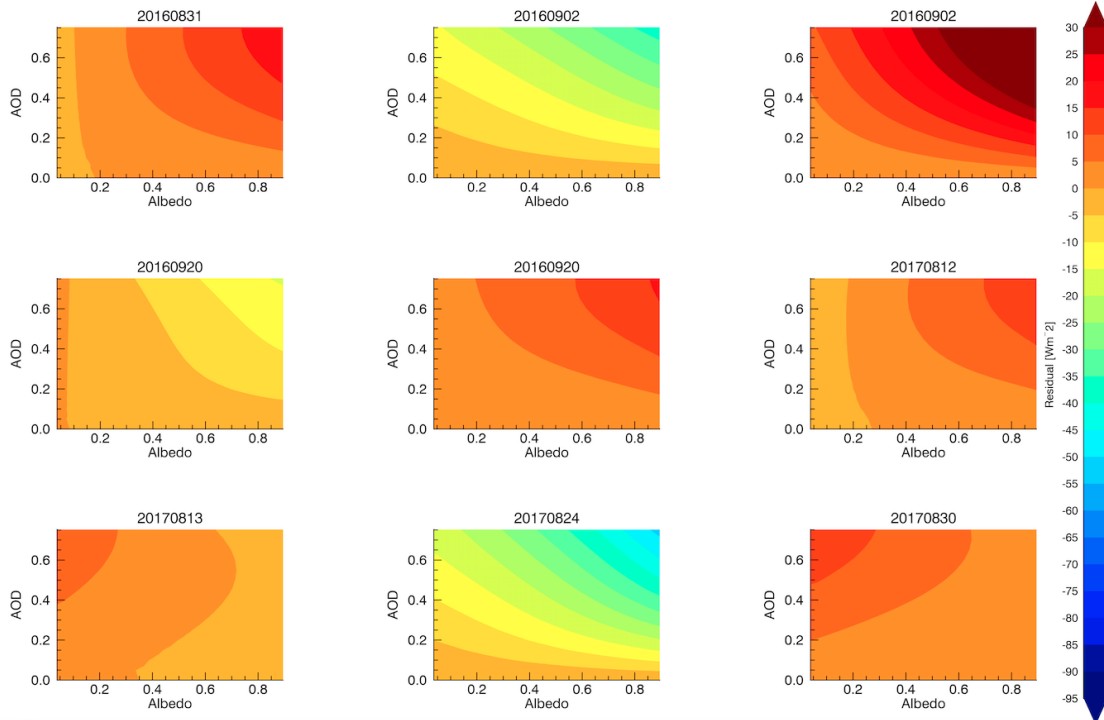

**Figure D1. Residual plot of directly calculated DARE (RTM output) and predicted BB DARE values using $P_{DARE}$ at a fixed SZA (20°).**





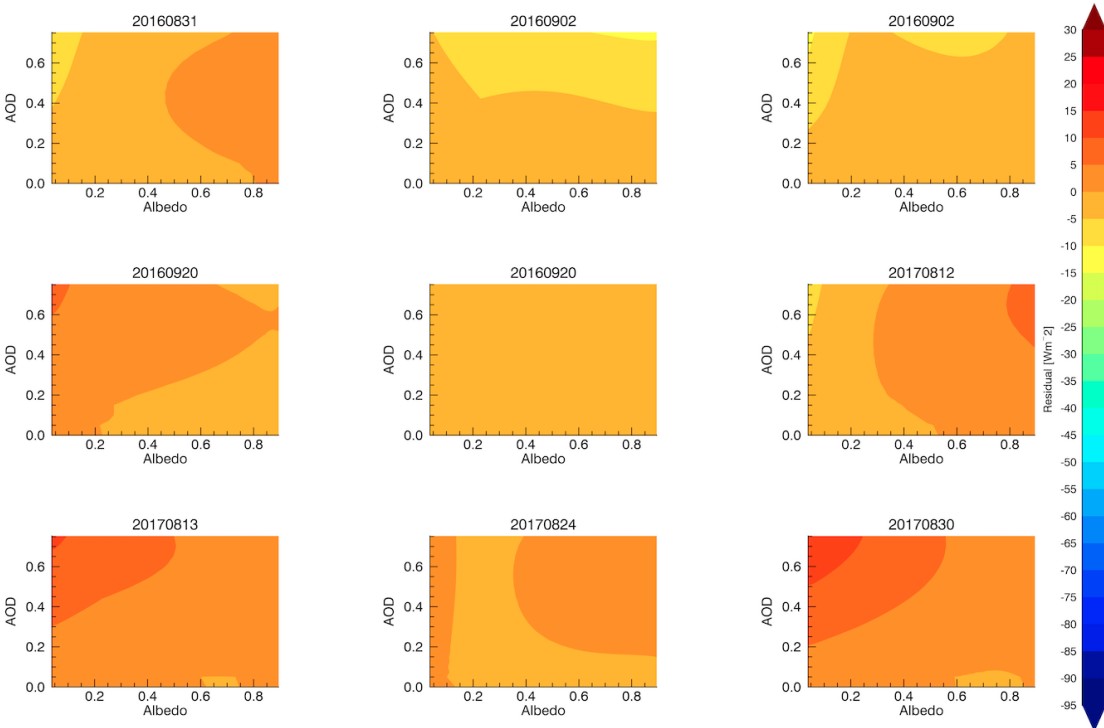

**Figure D2. Residual plots of directly calculated DARE (RTM output) and predicted broadband DARE values using PX$_{DARE}$ at a fixed SZA (20°).**

**Appendix E.**

Retrievals of SSA and *g* for each individual case with the associated retrieval uncertainty shown as error bars. Figure E1 shows the SSA retrievals for a) 2016 and b) 2017; E2 shows both the 2016 and 2017 *g* retrievals in one figure.

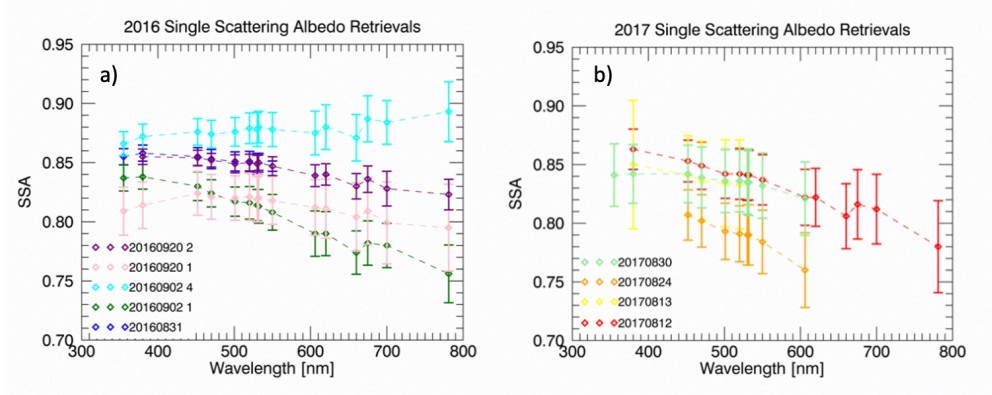

**Figure E1. SSA retrievals from a) 2016 and b) 2017 with associated retrieval uncertainty.**