# Peer review of "Empirically-Derived Parameterizations of the Direct Aerosol Radiative Effect based on ORACLES Aircraft Observations"

_Atmospheric Measurement Techniques, 2020_

## Referee Comment (RC1) · Jim Haywood (Referee) · 7 Jul 2020

Review of Cochrane et al, AMT

This paper builds on the previous work of Cochrane et al. (2019) and examines the direct radiative effect of biomass burning aerosols over cloudy and clear-sky conditions during the ORACLES 2016 and 2017 measurement campaigns. There are three main aspects: 1) assessing the aerosol optical parameters that give consistent radiative closure, 2) developing a basic and extended algorithm that parameterizes the DARE as a function of the scene albedo and the optical depth (at 550 nm) and 3) adding an additional dependency on SSA.

I was very interested and impressed by the SSA retrievals when compared to some of the more accurate assessments of SSA that are now possible using more advanced instrumentation (e.g. airborne CRD and PAS measurements that are not subject to the artefacts associated with missing scattering (nephelometers) or with scattering/absorption artefacts (filter based absorption measurements). I have included some additional references to these measurements as they were not included by the authors – I do think that these complementary studies provide excellent additional supporting information for the validity of the approach.

The second part of the paper which documents the performance of the P_DARE and PX_DARE could certainly be of use when compared to satellite data, but one would have to have an estimate of the above cloud/above surface AOD and the scene albedo. The most obvious place where this could have applicability might therefore be a combination of e.g. space borne lidar and broadband scene albedo from e.g. GERB; it might be worth explicitly stating this as a future possibility.

I deal with the comments as more major and minor below.

**More Major:**

Point 1: L325 onwards: It is worth emphasizing that many of the in-situ retrievals of SSA have, in the past and in the ORACLES measurements, relied on filter based measurements for absorption and nephelometers for scattering (authors' Figure 4). These instruments have relatively large uncertainties associated with them because of corrections needed for scattering and absorbing artefacts (e.g. Cappa et al., 2008). Much more accurate measurement systems such as Cavity Ring Down (extinction) and Photo Acoustic Spectrometry have been developed.

Cappa, C. D., Lack, D. A., Burkholder, J. B., and Ravishankara, A. R.: Bias in filter-based aerosol light absorption measurements due to organic aerosol loading: Evidence from laboratory measurements, Aerosol Sci. Technol., 42, 1022–1032, https://doi.org/10.1080/02786820802389285, 2008.

I know that it is difficult to keep up with the contemporary literature, particularly with the concentrated efforts over the SE Atlantic region, but there is some recent work from the CLARIFY-2017 team that very much supports the values of the SSA (and the wavelength dependence). I would suggest adding something like this:-

"New, more accurate, cavity ring down and photo acoustic spectrometry instrumentation has recently been deployed to the SE Atlantic during the CLARIFY-2017 deployment. Davies *et al*. (2019) performed an analysis of the SSA of aerosol dominated by biomass burning aerosol using such instrumentation and found mean SSA values of 0.84, 0.83 and 0.81 at interpolated wavelengths of 467, 528, and 652 nm respectively. Wu *et al*. (2020a) extended this analysis by examining the BBA in the free troposphere, finding a mean and variability in BBA SSA of 0.85 ± 0.02 (1stdev) and 0.82±0.04 (1stdev) at 405 and 658 nm with evidence that the BBA at higher altitudes in the free troposphere is less absorbing. These results appear entirely consistent with those derived here."

The authors should consider including a combination of the Davies et al. (2019)/Wu et al. (2020) paper on their Figure 4 as the agreement is so good…..Of course, you would have to caveat this with the fact that there are different temporal and geographical sampling regions etc.

Point 2: Line 234. The RT calculations themselves have a degree of uncertainty associated with them. For example, I note that the aerosol is characterized by the asymmetry factor. Does this mean that the higher order moments of the phase function are not accounted for? Is the RT code 2-stream? Is delta-Eddington rescaling applied? How is the surface reflectance modelled? A few more details would be appropriate here as would some acknowledgement that radiative transfer models that treat aerosols have their own inherent uncertainty (e.g. Boucher et al., 1999).

Point 3: The authors chose a SZA of 20degrees to demonstrate the RT calculations and the parameterization fits. As demonstration purposes, this is OK, but there should be an acknowledgement that, when comparing to models, the DRE is typically calculated over the full range of SZAs that are experienced in the region and then diurnally averaged.

**Minor typos/clarifications**

L24: spanned -> determined

L38: just off -> off (as the Sc extends 1000km….)

L48-49: "In a region like the southeast Atlantic, this makes determining DARE challenging since the cloud fields change rapidly". I would suggest adding some idea of why the cloud changes rapidly - not only because of cloud dynamics, but because the cloud field advection is dominated by the flow in the MBL while the aerosol advection is dominated by the flow in the residual continental marine boundary layer which is frequently in the opposite direction.

L52: Chand was not the first to coin the phrase critical surface albedo. I'd suggest adding Haywood and Shine, 1995 reference here (Haywood, J.M., and Shine, K.P., 1995. The effect of anthropogenic sulfate and soot aerosol on the clear sky planetary radiation budget. Geophys. Res. Letts., 22, 5, 603-606; see their Fig 1).

L75 (and probably other instances) aerosol optical depth -> AOD as you've already defined it

I like Figure 1. It captures the essence of the filters and the criteria.

L148: Figure 3a -> Figure 2a.

Caption for Figure 2. "c) The ratio between the BOL and TOL albedo spectra shown against the BOL AOD spectrum." I think that this needs a little more explanation. Presumably the AOD is 532nm? What about the albedo – is this the broadband albedo (i.e. weighted by the solar flux)?

L163-L164: "The filter, which is applied to the upwelling profile, retains only those data within the 68% confidence interval (1 sigma) of the linear fit line". This is fine if the error distribution is Gaussian, but is it? It would be worth checking that this is the case. It seems as though the two cases where the original filtering method is retained may not be Gaussian which might give you a reason for applying a different filtering method.

Line 255. I was initially concerned that the retrieval algorithms rely on a mean solar zenith angle. The spirals typically take 30mins to achieve. This means that the SZA could be 15minutes out at the TOL and BOL. I reckon that 15minutes is approximately equivalent to 3.75degrees error is SZA given that

the sza changes by around 15degrees per hour. So assuming a mean SZA of 30degrees (Table 2) could give an error of around 4% in the fluxes, or around 40Wm-2 assuming that you'd get around 1000Wm-2 from the product of the solar constant and atmospheric transmission. However, I note from an earlier section (line 152) that the fluxes are corrected according to Equation 3 of Cochrane et al. (2019); which is quite easy to miss on a first read. This therefore really needs to be reiterated here so emphasize that the observations are corrected and the observations and the modelling are therefore consistent.

L269: "We focus on the TOL calculations since radiative effects can be directly related to radiative balance at the TOL (Matus et al., 2015)." I think that it's better to say that the TOL calculations will resemble those calculated at the tropopause which is used as a metric for the cooling/warming impact of aerosols (e.g. Forster et al., 2007 which you already reference).

L281:   is DARE -> is the DARE

L286: Russel -> Russell; deGraaf -> de Graaf

L288: "no studies that we are aware of have generalized RFE to account for these complexities in a quantitative framework." Be careful here. General circulation modelling studies can (and do) turn out these numbers on a regular basis simply by taking the all sky DARE and dividing it by the AOD, which implicitly has all of the detailed RT calculations implicit within it. It is quoted numerous times for different aerosol types for a multitude of e.g. AEROCOM simulations. You could also argue that above cloud satellite retrieval estimates have also implicitly accounted for this in their look-up-tables (e.g. de Graaf et al., 2012, which you already cite).

What you mean is that "no detailed observational based studies"……

L295: Again, some care is needed here: "has the significant advantage that the complexities of transitioning from narrowband to broadband for many parameters are incorporated into the parameterization coefficients, allowing for use across large spatial scales since minimal information is required". If you have a different type of aerosol, or your aerosol is mixed with mineral dust (as it frequently is in west Africa), then your algorithm will fail because the above cloud AOD and the DARE will differ when compared to BBA alone. This note of caution needs to be included I think – easiest way is to tone down the "large" spatial scales, which is semi-quantitative to "regional".

L330: Russel -> Russell

L356: Might want to include reference to the Wu et al. (2020) paper here as that suggests that there is a variation in the vertical profile of SSA.

L360: (scene or cloud albedo). I'm a little confused – is the scene albedo, the albedo with the aerosol and the cloud in it, while the cloud albedo is the albedo of the cloud layer alone? The two must differ otherwise the aerosol is having no effect. A few words of clarification would be appropriate.

L382 and Figure 5. The authors have tended to slip into the terminology of "surface albedo" which tends to mean the physical reflectance of the surface rather than the "effective underlying albedo" (i.e. the albedo of the combined Rayleigh scattering, cloud, MBL aerosol and surface) which is I think what the authors mean. Again, this should be clarified. Is there an explicit assumption that the underlying albedo is Lambertian? While this might be a decent approximation for heterogeneous thick cloud, is it sufficient for the sea-surface reflectance? What impact would this have?

There are several figures where the albedo goes down to zero, but Figure 10 suggested a minimum surface albedo of around 3.5%, which is similar to a 'real' ocean surface – some more explanation is warranted I think.

L385-386. Fig 7 is referred to before Figure 6. Easy to sort out by swapping the order of this sentence.

L405: "and rather strongly on the SZA (not shown; for example, it can attain 0.6 at low Sun elevations)." I think it would be appropriate to include a reference here – the classical reference for the dependence of the DRE on the radiative forcing is Boucher et al (1999) although this is for sulfate aerosol.

One of the problems with presenting the critical SSA for a solar zenith angle of 20degrees is this rather strong solar zenith angle dependence. The values at 20degrees (close to local noon) is not likely to be representative of the mean SZA that is experienced in the region. A caveat to in this regard would be appropriate.

L 490: "We cannot judge whether our approach will be useful for predictive models, …... ". I agree that there are issues with whether the community will take up the parameterized approaches the RT calculations are a fundamental necessity. You might want to more explicitly suggest that a combination of lidar derived AODs and scene albedos from e.g. the geostationary GERB instrument or similar for future assessments of biomass burning DRE.

L494: "At the very least, the SSA and asymmetry parameter retrievals coming out of our and other ORACLES studies will constrain the aerosol optical properties in a range of models". Again I agree – the text and references that document the very encouraging agreement between the absolute values of the SSA and the spectral dependence of the SSA and those from higher accuracy CRD and PAS measurement systems coming out of CLARIFY-2017 should again be noted here I think. I would also suggest changing "ORACLES studies" to "ORACLES/LASIC/CLARIFY-2017/AEROCLO-Sa studies (Zuidema et al., 2016)".

References :
Boucher, O., et al., 1998. Intercomparison of models representing direct shortwave radiative forcing by sulphate aerosols. J. Geophys. Res., 103, 16979-16998.

Davies, N.W., C. Fox, K. Szpek, M.I. Cotterell, J.W. Taylor, J.D. Allan, P.I. Williams, J. Trembath, J.M. Haywood, and J.M. Langridge, Evaluating biases in filter-based aerosol absorption measurements using photoacoustic spectroscopy, Atmos. Meas. Tech., 12, 3417-3434, DOI: 10.5194/amt-12-3417-2019, 2019.

Wu, H.,  J.W. Taylor, K. Szpek, J. Langridge, P.I. Williams, M. Flynn, J.D. Allan, S.J. Abel, J. Pitt, M.I. Cotterell, C. Fox, N.W. Davies,  J. Haywood, H. Coe, Vertical and temporal variability of the properties of transported biomass burning aerosol over the southeast Atlantic during CLARIFY-2017, doi:10.5194/acp-2020-197, Atmos. Chem. Phys. Discuss., https://doi.org/10.5194/acp-2020-197, 2020.

Jim Haywood

---

## Referee Comment (RC2) · Anonymous Referee #2 · 10 Jul 2020

Review on "Empirically-Derived Parameterizations of the Direct Aerosol Radiative Effect based on ORACLES Aircraft Observations" by Cochrane et al.

Summary: This paper consists of two major parts. In the first part, an algorithm to retrieve the scattering properties of above-cloud smoke aerosols from air-borne SSFR measurements was introduced and applied to a few ORACLES cases. In the second part, the smoke aerosol scattering properties from the first part were first used to drive the radiative transfer model to compute the DARE which in turn were used to derive an empirical parameterization scheme of DARE.

The structure of this paper seems a little odd. It feels like two separate papers that deal

with very different topics being stitched together. The first part, i.e., aerosol retrieval algorithm, accounts for almost half of the length of the paper is not reflected in the title. There are a number of typos and small mistakes in the paper. In addition, the motivation for developing such DARE parameterization scheme is not clear to me. Overall, I think this manuscript needs significant revision and improvement before it can be accepted for publication. Below is a list of my questions and comments.

Major questions and comments: What is the motivation of this work? After reading the Introduction and even the whole paper, I'm still confused about the motivation of this paper. Why do the authors want to develop a DARE parameterization scheme? Note that the radiative transfer theory and methods for computing DARE have been well developed. There are also many broadband radiative transfer models (e.g., RRTM, Fu-Liou, Libradran) that are readily available. So, the computation of DARE is fairly straightforward once the aerosol scattering properties and the environmental factors (e.g., surface/cloud reflectance) are known. Why should someone use a less accurate and seemingly complicated parameterization scheme to estimate DARE?

What is the usefulness of the parameterization scheme and who should be interested in using it? The DARE computation and parameterization in this are based on the scattering properties retrieved from a handful research flights during the ORACLES campaign (i.e., SSA and g in Figure 3). While these measurements are unique and valuable, they are still highly limited in terms of sampling rate. Whether and how can the parameterization scheme be used to deal with the DARE computation in more general cases? For example, can it be used outside of ORACLES period or location when or where the aerosol scattering properties are different from those in in Figure 3? I think these questions should be clarified so that the readers can understand if and how the parameterization could be helpful for their research.

Significantly shorten the first part. As mentioned above, the structure of the paper feels odd. The first part, which is about 5 pages, describes a retrieval algorithm in detail, while the second part, which is about 6 pages, described a parameterization

scheme. The only connection between the two parts I can see is that the first part provides the scattering properties (i.e., SS and g) for the computation of DARE for the second part. In addition, the first part seems to be rather minor extension of a published method in Cochrane et al. (2019). Actually, it is very hard to follow the first part without reading the Cochrane et al. (2019) beforehand because of the frequent references it. So, it seems that the first part of the paper is not original or novel, and it only distracts the reader from the main point of this paper. For example, an understanding of how the SSFR measurements are filtered does not help the readers understand the DARE parameterization scheme at all. I would suggest shortening the first part substantially or putting most of it in the appendix to emphasize the most important and novel DARE parameterization part of the work.

Minor comments:

Line 24: Does the "scene albedo" actually mean cloud albedo? If it really means "scene albedo" then what types of scenes (ocean, land, snow etc.) have been included?

Line 86: similarly, it should be clarified here if "scene albedo" actually just means "cloud albedo". Note that the spectral signature of land reflectance/albedo is very different from cloud albedo.

Eq. (3) and (4): It should be pointed out explicitly if these equations are for instantaneous or diurnal averaged DARE. Also why is the dependence of DARE on solar zenith angle omitted in these equations? SZA is part of the parameterization (Table 4b), no?

To what extent is the DARE dependent on atmospheric profiles, such as water vapor profiles? There are some recent studies that suggest a correlation between the presence of above-cloud smoke and an enhanced water vapor in the ORACLE region. Should this correlation be considered in the parameterization?

Cochrane et al., 2019.has been cited many times in the paper, often in different formats. Please be consistent and also considering use abbreviation e.g., C19 to refer to

Cochrane et al., 2019.

Line 148: Figure 3 should be Figure 2.

Eq. (5) and (6), why are the parameters a_lambda and b_lambda the same for upward and downward irradiances? What is the underlying physics?

Again, Section 2.2.1 and 2.2.2 seem to be a replay of Cochrane et al., 2019. They should be put in the Appendix or substantially shortened.

Around line 150, this part is confusing and needs detail explanation. For example, "both upwelling (F_up) and downwelling (F_dn) irradiance profiles have an approximately linear relationship to AOD due to the absorption and scattering of the aerosol layer." Shouldn't the downwelling (F_dn) be exponential with AOD as a result of Beer's law? "Any deviation from the linear relationship is attributed to changes in the underlying cloud" Why? Can't the vertical variation of aerosol properties, e.g., SSA and/or g cause deviation from the linear relationship? How does cloud cause the deviation? These questions need to be clarified.

Does the retrieval algorithm assume H-G phase function? What are the higher-order terms of the phase function expansion other than asymmetry factor, g? What is the uncertainty associated with the phase function assumption?

Line 286: Russel et al., 1997 should be Russell et al., 1997; deGraaf should be de Graaf

Eq. (12) and (13): The formula looks quite arbitrary. Is there any physics behind these polynomial parametrizations or are they only empirical? Note that there are some well-established 2-stream or 4-stream formula for layer reflection, e.g., Coakley (1975). Is it possible to draw some theoretical basis or physical meaning for the parameterization from these 2-stream or 4-stream approximations? Also, some previous studies have tried to use the concept of adding doubling to approximate the reflection of two layers (Lenoble 1985). Do you think these formulae might be helpful?

Why is SZA dependence of DARE omitted in Eq. (12) and (13)?

Eq. (14) and (15): again, these parameterizations look arbitrary. Is there any underlying physics?

Eq. 21 – 24: I understand that dSSA term is introduced to make the parameterization scheme more general and more accurate. But as I mentioned above, a broadband RTM can easily compute the DARE given any type of SSA and g. Why bother developing such a complicated parameterization?

---

## Author Comment (AC1) · 16 Aug 2020

**Dear Jim Haywood,**

**Thank you for your positive review of our manuscript. We appreciate the comments and suggestions you provided and we believe they will greatly improve the clarity and scientific robustness of the manuscript. We would like to take this opportunity to provide you with our plan to address your main concerns of the manuscript *prior to submitting a revised manuscript* to allow for a continued discussion as necessary. Below we have provided responses (in boldface) to each of the major and minor comments.**

Review of Cochrane et al, AMT

This paper builds on the previous work of Cochrane et al. (2019) and examines the direct radiative effect of biomass burning aerosols over cloudy and clear-sky conditions during the ORACLES 2016 and 2017 measurement campaigns. There are three main aspects: 1) assessing the aerosol optical parameters that give consistent radiative closure, 2) developing a basic and extended algorithm that parameterizes the DARE as a function of the scene albedo and the optical depth (at 550 nm) and 3) adding an additional dependency on SSA.

I was very interested and impressed by the SSA retrievals when compared to some of the more accurate assessments of SSA that are now possible using more advanced instrumentation (e.g. airborne CRD and PAS measurements that are not subject to the artefacts associated with missing scattering (nephelometers) or with scattering/absorption artefacts (filter based absorption measurements). I have included some additional references to these measurements as they were not included by the authors – I do think that these complementary studies provide excellent additional supporting information for the validity of the approach.

The second part of the paper which documents the performance of the P_DARE and PX_DARE could certainly be of use when compared to satellite data, but one would have to have an estimate of the above cloud/above surface AOD and the scene albedo. The most obvious place where this could have applicability might therefore be a combination of e.g. space borne lidar and broadband scene albedo from e.g. GERB; it might be worth explicitly stating this as a future possibility.

I deal with the comments as more major and minor below.

**More Major:**

Point 1: L325 onwards: It is worth emphasizing that many of the in-situ retrievals of SSA have, in the past and in the ORACLES measurements, relied on filter based measurements for absorption and nephelometers for scattering (authors' Figure 4). These instruments have relatively large uncertainties associated with them because of corrections needed for scattering

and absorbing artefacts (e.g. Cappa et al., 2008). Much more accurate measurement systems such as Cavity Ring Down (extinction) and Photo Acoustic Spectrometry have been developed.

Cappa, C. D., Lack, D. A., Burkholder, J. B., and Ravishankara, A. R.: Bias in filter-based aerosol light absorption measurements due to organic aerosol loading: Evidence from laboratory measurements, Aerosol Sci. Technol., 42, 1022–1032, https://doi.org/10.1080/02786820802389285, 2008.

I know that it is difficult to keep up with the contemporary literature, particularly with the concentrated efforts over the SE Atlantic region, but there is some recent work from the CLARIFY- 2017 team that very much supports the values of the SSA (and the wavelength dependence). I would suggest adding something like this:-

"New, more accurate, cavity ring down and photo acoustic spectrometry instrumentation has recently been deployed to the SE Atlantic during the CLARIFY-2017 deployment. Davies *et al*. (2019) performed an analysis of the SSA of aerosol dominated by biomass burning aerosol using such instrumentation and found mean SSA values of 0.84, 0.83 and 0.81 at interpolated wavelengths of 467, 528, and 652 nm respectively. Wu *et al*. (2020a) extended this analysis by examining the BBA in the free troposphere, finding a mean and variability in BBA SSA of 0.85 ± 0.02 (1stdev) and 0.82±0.04 (1stdev) at 405 and 658 nm with evidence that the BBA at higher altitudes in the free troposphere is less absorbing. These results appear entirely consistent with those derived here."

The authors should consider including a combination of the Davies et al. (2019)/Wu et al. (2020) paper on their Figure 4 as the agreement is so good.....Of course, you would have to caveat this with the fact that there are different temporal and geographical sampling regions etc.

**Thank you for this comment and we apologize for our unintentional exclusion of the other SEA campaigns beyond ORACLES. We have included the suggested text on line 360. In figure 4, we will add an additional indicator of the Davies et al., 2019 results. We choose not to include the Wu et al., 2020 results since the wavelengths are not as similar.**

[Figure]

[Figure]

[Figure]

Point 2: Line 234. The RT calculations themselves have a degree of uncertainty associated with them. For example, I note that the aerosol is characterized by the asymmetry factor. Does this mean that the higher order moments of the phase function are not accounted for? Is the RT code 2-stream? Is delta-Eddington rescaling applied? How is the surface reflectance modelled? A few more details would be appropriate here as would some acknowledgement that radiative transfer models that treat aerosols have their own inherent uncertainty (e.g. Boucher et al., 1999).

**Thank you for pointing out the lack of detail surrounding the RT calculations within the manuscript. The RT calculations are performed with a multi stream RT model, disort, run with 6 streams. There is no delta-Eddington rescaling. The model assumes a Henyey-Greenstein (HG) phase function with our retrieved asymmetry parameter as input for the DARE calculations. We do not at all claim here that HG is, in fact, the actual phase function. On the contrary, the real phase function most likely significantly deviates from HG. However, since we retrieve g from _irradiance_ observations, using disort, and then we use this same disort to calculate DARE, HG(g) _represents_ the real phase function sufficiently well for our purposes. As you know, the first moment of the phase function along with a parametric HG phase function is sufficient to calculate fluxes with a two-stream approximation, whereas higher moments are required when using higher-stream RT models (including disort). The higher moments of the phase function can be generated from g (higher powers of it. For example, the second moment is simply $g^2$). The retrieval uncertainty in g (provided in the manuscript) could be propagated into higher moments, but since they are derived from g for HG, this is not necessary here.**

**For the aerosol retrievals and DARE RT calculations, we simply set the "surface" to the level of the cloud and define the albedo as the spectral cloud albedo. Therefore, we do not model the surface in the main calculations within the manuscript. However, for the DARE parameterization, the spectral cloud albedo "grid" at each SZA was obtained through cloud optical thickness and effective radius retrievals.**

**The translation from the mid-visible albedo to the spectral albedo (described in Appendix A.3.2) starts with the actually measured albedo spectrum. From that, COD and REF are retrieved, from which the spectrum is calculated. For the other albedos in the "grid", the COD is simply varied such that the albedo at the mid-visible wavelength changes as needed (while changing all of the other wavelengths as well).The COT/REF retrieval uses two wavelengths (860 and 1630 nm) to construct the lookup table, with no aerosol since the retrieval is done on the below aerosol, above cloud measurements. In these calculations, the surface for the cloud retrievals is standard Lambertian with an albedo value of 0.03. The COT range begins at 0, and this translates to a 0 "surface" albedo for the parameterization.**

**To address this comment in the manuscript, we will add the following information to this section in the paper:**

**Line 242: The RTM is run with 6 streams, assumes a Henyey Greenstein Phase function, and no delta-Eddington scaling is applied, all of which contribute to the inherent uncertainty within the RTM (Boucher et al., 1999).**

**Line 252: the RTM ingests the spectral cloud top albedo from SSFR (set at the altitude of the above-cloud leg, around 2km)**

**We should add that these simplifications were made to combine as many legs (cases) as possible in a common framework. In terms of radiative transfer, an opaque cloud effectively replaces the surface as boundary condition. However, for partially transparent clouds, the surface contributes to the reflectance. On the other hand, DARE is the *difference* between the fluxes with/without the aerosol above the altitude at which the albedo is prescribed in the model, and the approximations we used should therefore be insignificant compared to other uncertainties. Finally, it is acknowledged that clouds do not exhibit a Lambertian albedo. However, for irradiance calculations, the cloud albedo (non-Lambertian) can be substituted with a Lambertian albedo. We will add a brief clarification on these subtleties in the revised manuscript, provided our explanations here will be acceptable as path forward.**

Point 3: The authors chose a SZA of 20degrees to demonstrate the RT calculations and the parameterization fits. As demonstration purposes, this is OK, but there should be an acknowledgement that, when comparing to models, the DRE is typically calculated over the full range of SZAs that are experienced in the region and then diurnally averaged.

**Thank you for bringing this up. To make it clear that our parameterizations are instantaneous, the sentence preceding eq 3 and eq 4 will be changed from "ORACLES measurements are used collectively to develop two parameterizations in the form of:" to "ORACLES measurements are used collectively to develop two parameterizations of instantaneous DARE in the form of:" In addition, we will add the caveat to figure 5: "It should be noted that a 20° SZA is not representative of the mean in the region." We will further add that in the first application of our parameterization, the diurnal integration will be done (see also our response regarding applications below).**

**Minor typos/clarifications**

L24: spanned -> determined

**The abstract has been rewritten (below, in blue) to better reflect the main goals of the paper. This wording no longer remains.**

Revised Abstract: In this manuscript, we use observations from the NASA ORACLES (ObseRvations of CLouds above Aerosols and their intEractionS) aircraft campaign to develop a framework by way of two parameterizations that establishes regionally representative relationships between aerosol-cloud properties and their radiative effects. These relationships rely on new spectral aerosol property retrievals of the single scattering albedo (SSA) and

asymmetry parameter (ASY).  The retrievals capture the natural variability of the study region as sampled, and both were found to be fairly narrowly constrained (SSA: 0.83 ± 0.03 in the mid-visible, 532 nm; ASY: 0.54 ± 0.06 at 532 nm). The spectral retrievals are well suited to calculate the direct aerosol radiative effect (DARE) since SSA and ASY are tied directly to the irradiance measured in presence of aerosols – one of the inputs to the spectral DARE.

The framework allows for entire campaigns to be generalized into a set of parameterizations. For a range of solar zenith angles, it links the broadband DARE to the mid-visible aerosol optical thickness (AOD) $\tau$ and the albedo $\alpha$ of the underlying scene (either clouds or clear sky) by way of the first parameterization: P($\tau$, $\alpha$). For ORACLES, the majority of the case-to-case variability of the broadband DARE is attributable to the dependence on the two driving parameters of P($\tau$, $\alpha$). A second, extended, parameterization PX($\tau$, $\alpha$, $\varpi$) explains even more of the case-to-case variability by introducing the mid-visible SSA $\varpi$ as third parameter. These parameterizations establish a direct link from two or three mid-visible (narrowband) parameters to the broadband DARE, implicitly accounting for the underlying spectral dependencies of its drivers. They circumvent some of the assumptions when calculating DARE from satellite products, or in a modeling context. For example, the DARE dependence on aerosol microphysical properties is not explicit in P or PX because the asymmetry parameter varies too little from case to case to translate into appreciable DARE variability. While these particular DARE parameterizations only represent the ORACLES data, they raise the prospect of generalizing the framework to other regions.

L38: just off -> off (as the Sc extends 1000km....)

**This will be adjusted.**

L48-49: "In a region like the southeast Atlantic, this makes determining DARE challenging since the cloud fields change rapidly". I would suggest adding some idea of why the cloud changes rapidly - not only because of cloud dynamics, but because the cloud field advection is dominated by the flow in the MBL while the aerosol advection is dominated by the flow in the residual continental marine boundary layer which is frequently in the opposite direction.

**Thank you, additional details will be included here.**

L52: Chand was not the first to coin the phrase critical surface albedo. I'd suggest adding Haywood and Shine, 1995 reference here (Haywood, J.M., and Shine, K.P., 1995. The effect of anthropogenic sulfate and soot aerosol on the clear sky planetary radiation budget. Geophys. Res. Letts., 22, 5, 603- 606; see their Fig 1).

**Reference will be added. Thank you for pointing out our oversight.**

L75 (and probably other instances) aerosol optical depth -> AOD as you've already defined it

**Thank you, this will be updated.**

I like Figure 1. It captures the essence of the filters and the criteria.

**Thank you.**

L148: Figure 3a -> Figure 2a.

**Figure reference will be updated.**

Caption for Figure 2. "c) The ratio between the BOL and TOL albedo spectra shown against the BOL AOD spectrum." I think that this needs a little more explanation. Presumably the AOD is 532nm? What about the albedo – is this the broadband albedo (i.e. weighted by the solar flux)?

**Figure 2c shows the ratio between the two albedo spectra from 2b (BOL and TOL SSFR spectral albedo) as a function of the 4STAR AOD spectrum at the BOL. To make this more clear, the figure caption will be updated to: "c) The ratio between the BOL and TOL albedo spectra (taken from Fig 2b) shown against the BOL AOD spectrum at the 4STAR wavelengths. The intercept of the fit line is criteria 3 ($AR_\infty$); if the intercept deviates largely from 1.0, the case cannot be used for an aerosol retrieval. Select wavelengths are labelled to highlight the spectral importance of this method."**

**In addition, we will update the figure to highlight a few wavelengths so it is clear that the data shown is spectral. Below is the proposed figure:**

[Figure]

L163-L164: "The filter, which is applied to the upwelling profile, retains only those data within the 68% confidence interval (1 sigma) of the linear fit line". This is fine if the error distribution is Gaussian, but is it? It would be worth checking that this is the case. It seems as though the two cases where the original filtering method is retained may not be Gaussian which might give you a reason for applying a different filtering method.

**Technically, the measurement data should be Gaussian, since Gaussian is essentially measurement repeatability with some variability. For the stationary condition, if we are measuring over the same surface the measurements should be Gaussian distributed, or if noise is present, that should be Gaussian distributed. However, if the cloud changes within the measurement period, one no longer measures the same thing, and that will be reflected in the data. The idea is that through the filtering, we get rid of anything that falls outside of that Gaussian envelope since the data become too unlikely.**

In the original filtering method (from Cochrane et al., 2019), the 1-sigma standard deviation limit is based on the mean value of the profile. The updated filtering method uses the 1-sigma standard deviation based on the *linear fit* of the profile. In most cases, the new filtering provided better outcomes for the quality criteria (C1, C2, C3 in Fig 1) that follow the filtering, indicating that the new filter was eliminating more of the outlying data and retaining the quality data.  However, in the exception cases, we found the opposite and the new filter resulted in poorer values of the quality criteria. In these cases, we reverted back to the old filter method.

Below are the comparisons between the new and old filtering below (shown for the two exception cases that pass the criteria for a retrieval and one for a standard case that uses the new filtering). The cyan color is data within the 1 sigma fit and therefore retained; the lighter colors (pink, orange, purple) are data that do not pass the filter and not retained. In the first case, the new filter eliminates the highest AOD data. In the second case, the new filter retains too much variability at the highest AOD (lowest altitude).

Case 1:

[Figure]

| Original fit | New fit |

Case 2:

[Figure]

| Original fit | New fit |

In most cases, the difference between the two filtering methods is very small. Below is an example of a standard case for which the new fit is used.

[Figure]

| Original fit | New fit |

Line 255. I was initially concerned that the retrieval algorithms rely on a mean solar zenith angle. The spirals typically take 30mins to achieve. This means that the SZA could be 15minutes out at the TOL and BOL. I reckon that 15minutes is approximately equivalent to 3.75degrees error is SZA given that the sza changes by around 15degrees per hour. So assuming a mean SZA of 30degrees (Table 2) could give an error of around 4% in the fluxes, or around 40Wm-2 assuming that you'd get around 1000Wm- 2 from the product of the solar constant and atmospheric transmission. However, I note from an earlier section (line 152) that the fluxes are

corrected according to Equation 3 of Cochrane et al. (2019); which is quite easy to miss on a first read. This therefore really needs to be reiterated here so emphasize that the observations are corrected and the observations and the modelling are therefore consistent.

**We will amend this to: "As mentioned in section 2.2.1, the irradiances are corrected to the SZA at the midpoint of the spiral to account for the changing solar position during the spiral. For consistency, the SZA within the RTM is set to the same SZA of the spiral midpoint."**

L269: "We focus on the TOL calculations since radiative effects can be directly related to radiative balance at the TOL (Matus et al., 2015)." I think that it's better to say that the TOL calculations will resemble those calculated at the tropopause which is used as a metric for the cooling/warming impact of aerosols (e.g. Forster et al., 2007 which you already reference).

**Thank you for pointing this out. Indeed, our TOL results should certainly resemble the top of the troposphere more than the entire column. This will be updated.**

L281: is DARE -> is the DARE

**This will be updated.**

L286: Russel -> Russell; deGraaf -> de Graaf

**Thank you for catching this error, it will be corrected.**

L288: "no studies that we are aware of have generalized RFE to account for these complexities in a quantitative framework." Be careful here. General circulation modelling studies can (and do) turn out these numbers on a regular basis simply by taking the all sky DARE and dividing it by the AOD, which implicitly has all of the detailed RT calculations implicit within it. It is quoted numerous times for different aerosol types for a multitude of e.g. AEROCOM simulations. You could also argue that above cloud satellite retrieval estimates have also implicitly accounted for this in their look-up-tables (e.g. de Graaf et al., 2012, which you already cite). What you mean is that "no detailed observational based studies"......

**Based on your comment, it is clear that our intended meaning is not expressed through this statement and we therefore have decided to remove it from the manuscript. Rather than the RFE itself, this statement was intended to pertain to the parameterization, going from a single-parameter concept (RFE, used with AOD) to multiple parameters (as the ones driving P/PX, i.e., AOD, albedo, SSA).**

L295: Again, some care is needed here: "has the significant advantage that the complexities of transitioning from narrowband to broadband for many parameters are incorporated into the parameterization coefficients, allowing for use across large spatial scales since minimal information is required". If you have a different type of aerosol, or your aerosol is mixed with

mineral dust (as it frequently is in west Africa), then your algorithm will fail because the above cloud AOD and the DARE will differ when compared to BBA alone. This note of caution needs to be included I think – easiest way is to tone down the "large" spatial scales, which is semi-quantitative to "regional".

**We agree that caution is needed in the application of the parameterization to larger regions, since with a different aerosol type the coefficients will no longer be valid. We will make this more clear by changing the text to: "has the significant advantage that the complexities of transitioning from narrowband to broadband for many parameters are incorporated into the parameterization coefficients, allowing for use across regional spatial scales for biomass burning aerosol since minimal information is required as input. Of course, the parameterization is only applicable for the region where the measurements were taken. It also cannot be generalized to apply for a different aerosol type."**

L330: Russel -> Russell

**This will be updated.**

L356: Might want to include reference to the Wu et al. (2020) paper here as that suggests that there is a variation in the vertical profile of SSA.

**Reference will be added, thank you.**

L360: (scene or cloud albedo). I'm a little confused – is the scene albedo, the albedo with the aerosol and the cloud in it, while the cloud albedo is the albedo of the cloud layer alone? The two must differ otherwise the aerosol is having no effect. A few words of clarification would be appropriate.

**When referring to scene albedo, we mean that as the scene *below* the aerosol layer. This work, especially in relation to the parameterizations, does not focus on the albedo measured from above the aerosol layer. For more clarity, line 86 in the paper will be changed from "The 550 nm albedo is the albedo of the scene below the aerosol layer and the SSA is a measure of aerosol absorption." to "The 550 nm albedo is the albedo of the scene below the aerosol layer (open ocean and/or cloudy scene), and the SSA is a measure of aerosol absorption."**

**We differentiate between scene and cloud in this instance because the parameterizations include albedo values down to 0. Again for more clarity, we will amend this phrase to: (scene or cloud albedo below the aerosol layer). See also our response above how cloud-free scenes are handled in the radiative transfer.**

L382 and Figure 5. The authors have tended to slip into the terminology of "surface albedo" which tends to mean the physical reflectance of the surface rather than the "effective underlying albedo" (i.e. the albedo of the combined Rayleigh scattering, cloud, MBL aerosol and surface) which is I think what the authors mean. Again, this should be clarified. Is there an

explicit assumption that the underlying albedo is Lambertian? While this might be a decent approximation for heterogeneous thick cloud, is it sufficient for the sea-surface reflectance? What impact would this have?

There are several figures where the albedo goes down to zero, but Figure 10 suggested a minimum surface albedo of around 3.5%, which is similar to a 'real' ocean surface – some more explanation is warranted I think.

**This is a very good point. All mentions of surface albedo in the manuscript will be changed to underlying albedo.**

**All of the albedo spectra in the paper are at cloud level, around 2 km, so they do include Rayleigh scattering. We realize Rayleigh scattering will change with altitude, but within the parameterization that change will be small (since the altitude variation between cases is small) and only would matter for the albedo translation. Within the radiative transfer calculations (as described under response to main point #2), the 'clear sky' 0 albedo is the albedo in the limit of 0 cloud optical thickness, and not a true clear sky with a 0 surface albedo. There is still a cloud with an optical thickness of 0 and Lambertian assumption.**

**Also, it is acknowledged that a sea surface is even less of a Lambertian reflector than a cloud (one only needs to think of sun glint). However, this is precisely the simplification that we made to fit both cloudy and cloud-free skies into a common framework. In a sense, one could call our albedo an "effective" albedo, which represents the true surface reflectance assuming a Lambertian reflectance distribution. We will add a statement in the manuscript about this point.**

**Finally, since we are looking at DARE (the difference of fluxes) rather than the fluxes themselves, our simplifications should lead to only negligible effects relative to the contributing measurement uncertainties.**

L385-386. Fig 7 is referred to before Figure 6. Easy to sort out by swapping the order of this sentence.

**The sentence order will be switched.**

L405: "and rather strongly on the SZA (not shown; for example, it can attain 0.6 at low Sun elevations)." I think it would be appropriate to include a reference here – the classical reference for the dependence of the DRE on the radiative forcing is Boucher et al (1999) although this is for sulfate aerosol.

**Thank you, reference will be added.**

One of the problems with presenting the critical SSA for a solar zenith angle of 20degrees is this rather strong solar zenith angle dependence. The values at 20degrees (close to local noon) is

not likely to be representative of the mean SZA that is experienced in the region. A caveat to in this regard would be appropriate.

**Thank you, we will add in this caveat into the caption of figure 5 (since this is the first figure shown at 20 degree SZA.) For your reference, below are the critical SSA figures for SZA=20 and SZA=40:**

[Figure]

L 490: "We cannot judge whether our approach will be useful for predictive models, ...... ". I agree that there are issues with whether the community will take up the parameterized approaches the RT calculations are a fundamental necessity. You might want to more explicitly suggest that a combination of lidar derived AODs and scene albedos from e.g. the geostationary GERB instrument or similar for future assessments of biomass burning DRE.

**Thank you for this suggestion. In fact, we are working on an application paper that does just what you are suggesting here! We will include the additional statement: "However, current work implementing our approach using albedo data from the geostationary Spinning Enhanced Visible and Infrared Imager (SEVIRI) in combination with ORACLES AOD data from HSRL-2 and 4STAR is already underway (Chen et al., 2020 in preparation)."**

L494: "At the very least, the SSA and asymmetry parameter retrievals coming out of our and other ORACLES studies will constrain the aerosol optical properties in a range of models". Again I agree – the text and references that document the very encouraging agreement between the absolute values of the SSA and the spectral dependence of the SSA and those from higher accuracy CRD and PAS measurement systems coming out of CLARIFY-2017 should again be noted here I think. I would also suggest changing "ORACLES studies" to "ORACLES/LASIC/CLARIFY-2017/AEROCLO-Sa studies (Zuidema et al., 2016)".

**We agree that this should be highlighted. The text will be updated to: "At the very least, the agreement between the absolute values and spectral dependence of the SSA and asymmetry parameter retrievals coming out of our and other ORACLES/LASIC/CLARIFY-2017/AEROCLO-Sa studies (Zuidema et al., 2016) such as Davies et al. (2019) and Wu et al. (2020) will provide robust and consistent constraints of the aerosol optical properties in a range of models."**

References :

Boucher, O., et al., 1998. Intercomparison of models representing direct shortwave radiative forcing by sulphate aerosols. J. Geophys. Res., 103, 16979-16998.

Davies, N.W., C. Fox, K. Szpek, M.I. Cotterell, J.W. Taylor, J.D. Allan, P.I. Williams, J. Trembath, J.M. Haywood, and J.M. Langridge, Evaluating biases in filter-based aerosol absorption measurements using photoacoustic spectroscopy, Atmos. Meas. Tech., 12, 3417-3434, DOI: 10.5194/amt-12-3417- 2019, 2019.

Wu, H., J.W. Taylor, K. Szpek, J. Langridge, P.I. Williams, M. Flynn, J.D. Allan, S.J. Abel, J. Pitt, M.I. Cotterell, C. Fox, N.W. Davies, J. Haywood, H. Coe, Vertical and temporal variability of the properties of transported biomass burning aerosol over the southeast Atlantic during CLARIFY-2017, doi:10.5194/acp-2020-197, Atmos. Chem. Phys. Discuss., https://doi.org/10.5194/acp-2020-197, 2020.

---

## Author Comment (AC2) · 16 Aug 2020

Dear reviewer,

Thank you for your thoughtful and constructive review of the manuscript. We would like to take this opportunity to provide you with our plan to address your main concerns of the manuscript *prior to submitting a revised manuscript* to allow for a continued discussion as necessary. In the following paragraphs, we have summarized and responded to what we understand to be the main points of concern. We have also provided specific responses including plots to the individual minor comments. In case you take issue with our viewpoints or have any other recommendations, we'd appreciate if you could respond in this forum prior to us implementing the changes in the revised manuscript.

General comments/approach

It is evident from your comments that we are lacking clarity in the motivation of developing a DARE parameterization, which is key to the success of the paper. We appreciate this comment and acknowledge that the purpose was not made clear throughout the manuscript. The major advantage of developing the DARE parameterization is to encapsulate the transition from narrowband to broadband such that spectral aerosol and cloud properties are not required. The parameterizations require only one wavelength of input parameters and provides output of broadband DARE. Our goal was not to create an alternative to well-established radiative transfer formulae such as those presented in Coakley 1975, nor to replace full RT models such as libRadtran (with its solver "disort" created by Warren Wiscombe). Rather, the difficulty concerns the *spectral* aerosol and cloud properties and their relationship to the *broadband* DARE. One cannot simply translate SSA, g, cloud albedo, etc. at a single wavelength into broadband DARE since their spectral dependencies are almost as important as the narrow-band single-wavelength values. Established formulae that are grounded in the physics (such as Coakley's or other 2-stream approximations) still require spectral properties and subsequent broadband integration. Here, we replace the need for a full spectrum with narrow-band quantities at a wavelength in the mid-visible wavelength range where they are often available, and we also side-step the spectral integration. This, rather than the radiative transfer itself, is the essence of this work. More generally, we cast the collective radiative observations into an ORACLES-wide set of formulae and "resolve" the case-to-case DARE variability in terms of only two (or three) driving parameters. To express this better, we already re-wrote the abstract of the paper (included in blue font below). Again, we'd appreciate if you could let us know whether this explains the intent of the paper better (or more importantly, if not):

Revised Abstract: In this manuscript, we use observations from the NASA ORACLES (ObseRvations of CLouds above Aerosols and their intEractionS) aircraft campaign to develop a framework by way of two parameterizations that establishes regionally representative relationships between aerosol-cloud properties and their radiative effects. These relationships rely on new spectral aerosol property retrievals of the single scattering albedo (SSA) and asymmetry parameter (ASY). The retrievals capture the natural variability of the study region as sampled, and both were found to be fairly narrowly constrained (SSA: 0.83 ± 0.03 in the mid-visible, 532 nm; ASY: 0.54 ± 0.06 at 532 nm). The spectral retrievals are well suited to calculate the direct aerosol radiative effect (DARE) since SSA and ASY are tied directly to the irradiance measured in presence of aerosols – one of the inputs to the spectral DARE.

The framework allows for entire campaigns to be generalized into a set of parameterizations. For a range of solar zenith angles, it links the broadband DARE to the mid-visible aerosol optical thickness (AOD) $\tau$ and the albedo $\alpha$ of the underlying scene (either clouds or clear sky) by way of the first parameterization: $P(\tau, \alpha)$. For ORACLES, the majority of the case-to-case variability of the broadband DARE is attributable to the dependence on the two driving parameters of $P(\tau, \alpha)$. A second, extended, parameterization $PX(\tau, \alpha, \varpi)$ explains even more of the case-to-case variability by introducing the mid-visible SSA $\varpi$ as third parameter. These parameterizations establish a direct link from two or three mid-visible (narrowband) parameters to the broadband DARE, implicitly accounting for the underlying spectral dependencies of its drivers. They circumvent some of the assumptions when calculating DARE from satellite products, or in a modeling context. For example, the DARE dependence on aerosol microphysical properties is not explicit in P or PX because the asymmetry parameter varies too little from case to case to translate into appreciable DARE variability. While these particular DARE parameterizations only represent the ORACLES data, they raise the prospect of generalizing the framework to other regions.

The *application* of the parameterizations will be made clearer in the revised manuscript, with the important addition of explicitly stating how this is already being used to obtain diurnally-integrated values of DARE for the study region. We are actively working on a publication that uses the parameterization to establish diurnally-integrated DARE for the study region, based on satellite observations and aircraft observations of AOD and albedo. It is important to note, however, that the parameterizations were not developed for regions *beyond* that of ORACLES, since as you pointed out, the physics in other regions are most likely different and ours is an empirical formula that applies only to the data it is based on (we alluded to aerosol microphysics specifically in the revised abstract quoted above). While we recognize that the number of cases is small, the range of properties they cover is likely sufficiently representative of the conditions in the study region for developing the parameterization scheme. As more cases become available, even those falling outside of the envelope of, say, previously observed single scattering albedo, the *relationship between* mid-visible aerosol properties (AOD, SSA, asymmetry parameter), albedo on the one hand, and broadband DARE on the other would likely be similar as well. The advantage of our framework is that it can be continually updated. This will be done, for example, with the aforementioned paper that we are currently working on. Whereas the present paper only considers ORACLES 2016 and 2017, the follow-up paper extends the parameterization to ORACLES 2018, prior to the application of the parameterization to the calculation of campaign-averaged DARE. Data from other campaigns (e.g., from CLARIFY or even SAFARI) could similarly be included.

We understand that the two parts of the paper seem distinct from one another, and this is done purposefully. The manuscript is the second part to its companion paper, Cochrane et al., 2019, and they are meant to be read in succession. However, it is important that we include enough detail in the first part of the paper on the retrieval algorithm such that it can be understood at a surface level without reading the 2019 companion paper, as well as to make clear the changes made to the algorithm relative to the version described in the 2019 paper. The most important change of the 2020 version of the algorithm relative to its 2019 predecessor is that the data are automatically filtered and conditioned in a consistent manner. This allows ingesting of data without different treatment of cases and thereby biasing the results. In addition, part I of the current paper provides a data product on its own: the aerosol property retrievals of SSA and g. This database of these retrievals, though small, is important to include along with the parameterization of DARE as presented in part II.

Provided that this is an acceptable avenue for addressing your concerns, here is what we plan to do for the revised manuscript:

First, the abstract has been revised to better reflect the overall goals of the paper (see draft above in blue). As it currently reads, it does not effectively capture the main objectives of the paper and potentially leaves the reader with an incorrect impression of what the paper is about. Within the body of the manuscript, the transition from narrowband to broadband benefit of our parameterization will also be highlighted in greater detail. In addition, the application of our parameterizations will be more explicitly described (using the next paper as example). We will make the separation between part I and part II of the paper even more apparent, making it possible to skip the aerosol property retrieval section and move directly into the DARE section. However, we are hesitant to significantly reduce this section because of its value to the southeast Atlantic region research community.

Thank you again for your review of the paper and we hope that our changes will address your concerns satisfactorily.

With the above comments, we address the general (major) comments. Our response to sequential (minor) comments is below (in blue).

Minor comments:

Line 24: Does the "scene albedo" actually mean cloud albedo? If it really means "scene albedo" then what types of scenes (ocean, land, snow etc.) have been included?

ORACLES data was exclusively taken over the southeast Atlantic ocean. We use "scene albedo" rather than "cloud albedo" since we encountered scenes that included both cloudy and clear sky (partially cloudy or area contained cloud holes.) This will be clarified in the revised manuscript (it is already described in the newly drafted abstract quoted above).

Line 86: similarly, it should be clarified here if "scene albedo" actually just means "cloud albedo". Note that the spectral signature of land reflectance/albedo is very different from cloud albedo.

Thank you for pointing out this ambiguity. For more clarity, the line has been changed from "The 550 nm albedo is the albedo of the scene below the aerosol layer and the SSA is a measure of aerosol absorption." to "The 550 nm albedo is the albedo of the scene below the aerosol layer (open ocean and/or cloudy scene), and the SSA is a measure of aerosol absorption."

Eq. (3) and (4): It should be pointed out explicitly if these equations are for instantaneous or diurnal averaged DARE. Also why is the dependence of DARE on solar zenith angle omitted in these equations? SZA is part of the parameterization (Table 4b), no?

Thank you, the sentence preceding eq 3 and eq 4 has been changed from "ORACLES measurements are used collectively to develop two parameterizations in the form of:" to "ORACLES measurements are used collectively to develop two parameterizations of instantaneous DARE in the form of:"

The dependence of DARE on SZA is not included in these equations, since the parameterization coefficients themselves do not depend on SZA. Rather, the parameterization coefficients are calculated separately for a range of SZAs.

To what extent is the DARE dependent on atmospheric profiles, such as water vapor profiles? There are some recent studies that suggest a correlation between the presence of above-cloud smoke and an enhanced water vapor in the ORACLE region. Should this correlation be considered in the parameterization?

This is an important question. The dependence of DARE on atmospheric profiles, specifically water vapor, is implicitly included within the parameterization. Each case that the parameterization is based upon has the measured water vapor profile included in its DARE calculations.

While we do not determine the extent to which DARE depends on water vapor, it should not depend too much on it since DARE is defined as Fnet (with aerosol) – Fnet (without aerosol). The water vapor profile for both terms was held fixed (as determined from observations for each of the cases), and therefore only has a bearing on DARE if it affects the incident radiation on top of the aerosol layer. Compared to the impact of the scene AOD, albedo, and SSA on the case-to-case DARE variability, the water vapor profile variability (as the aerosol microphysics variability) was negligible. However, the water vapor does have a bearing on the total heating rates and is explored in a separate paper (in progress.)

Cochrane et al., 2019.has been cited many times in the paper, often in different formats. Please be consistent and also considering use abbreviation e.g., C19 to refer to Cochrane et al., 2019.

Thank you for catching this. The references to Cochrane et al., 2019 will be abbreviated to C19 (defined upon its first occurrence).

Line 148: Figure 3 should be Figure 2.

Thank you. This will be updated.

Eq. (5) and (6), why are the parameters a_lambda and b_lambda the same for upward and downward irradiances? What is the underlying physics?

Thank you for pointing this out, as our notation is ambiguous. $a_\lambda^\uparrow$ is not the same as $a_\lambda^\downarrow$ (similar for $b_\lambda^\uparrow$ and $b_\lambda^\downarrow$; equation 6 will be changed to $c_\lambda^\uparrow$ and $d_\lambda^\downarrow$.

Again, Section 2.2.1 and 2.2.2 seem to be a replay of Cochrane et al., 2019. They should be put in the Appendix or substantially shortened.

As noted in the response to the general comments, we understand that the two parts of the paper seem distinct from one another, and this is done purposefully. The manuscript is the second part to its companion paper, Cochrane et al., 2019, and they are meant to be read in succession. However, it is important that we include enough detail in the first part of the paper on the retrieval algorithm such that it can be understood at a surface level without reading the 2019 companion paper, as well as to make clear the changes made to the algorithm relative to the version described in the 2019 paper. The most important change of the 2020 version of the algorithm relative to its 2019 predecessor is that the data are automatically filtered and conditioned in a consistent manner. This allows ingesting of data without different treatment of cases and thereby biasing the results. In addition, part I of the current paper provides a data product on its own: the aerosol property retrievals of SSA and g. This database of these retrievals, though small, is important to include along with the parameterization of DARE as presented in part II.

Around line 150, this part is confusing and needs detail explanation. For example, "both upwelling ($F\_up$) and downwelling ($F\_dn$) irradiance profiles have an approximately linear relationship to AOD due to the absorption and scattering of the aerosol layer." Shouldn't the downwelling ($F\_dn$) be exponential with AOD as a result of Beer's law? "Any deviation from the linear relationship is attributed to changes in the underlying cloud" Why? Can't the vertical variation of aerosol properties, e.g., SSA and/or g cause deviation from the linear relationship? How does cloud cause the deviation? These questions need to be clarified.

We did not intend to imply that the direct downwelling irradiance should not follow Beer's law. However, in the SSFR measurements we cannot separate the global irradiance into its diffuse and direct components. We are using the linear assumption for the global downwelling as a simplification only for initial fitting for the subsequent filtering. We agree with the reviewer here because indeed, deviations from the linear relationship could be due to non-linearities as expected from Beer's law, or vertical dependencies of aerosol parameters. However, we expect these to be negligible compared to changes in the underlying clouds, and therefore use deviations from a linear profile to filter our data.

As can be seen from the figures below, the direct-beam downwelling (labeled SPN-f, SPN-S) does follow an exponential (these measurements were only available during 2018, and they are therefore not included in this manuscript). However, the global downwelling (SSFR global, green) can be approximated with a linear fit, at least for the range of AOD we encountered. We should add that the thin-layer approximation introduced by, e.g., Coakley et al. (1975) shows a linear dependence of the global reflectance analytically ("In the thin atmosphere approximation, the reflectivity and transmissivity of the layer are linear functions of its optical depth"). The figure below on the right shows the irradiance components as a function of altitude.

[Figure]

While vertical variation of the aerosol properties could theoretically affect the linear relationship that we assumed, but changes in the cloud properties below outweigh aerosol-induced deviations from a linear profile by far, and can therefore be used as the basis for filtering.

Does the retrieval algorithm assume H-G phase function? What are the higher-order terms of the phase function expansion other than asymmetry factor, g? What is the uncertainty associated with the phase function assumption?

Yes, it does assume an Henyey-Greenstein (HG) phase function with our retrieved asymmetry parameter (the first moment of the actual phase function) as input. The first moment of the phase function along with a parametric HG phase function is sufficient to calculate fluxes with a two-stream approximation. Higher moments are required when using higher-stream RT models (including disort) and can be generated from g (higher powers of it; for example, the second moment is simply $g^2$). We do not at all claim here that HG is, in fact, the actual phase function. On the contrary, the real phase function most likely deviates from HG. However, HG(g) *represents* the real phase function sufficiently well for our purposes. Using Coakley '75 again as an example: He does show that there are differences in the reflectance when using a two-stream formula relative to a multi-stream (in this case, doubling adding) RT. However, these differences arose because they were looking at directional reflectance. In our case, we retrieve g from *irradiance* observations, using disort, and then we use this same disort to calculate

DARE. The retrieval uncertainty in g (provided in the manuscript) could propagate into higher moments, but since they are trivially derived from g for HG ($g^2$, $g^3$, etc.), this is not necessary here.

Line 286: Russel et al., 1997 should be Russell et al., 1997; deGraaf should be de Graaf

Updated, thank you for pointing out this error.

Eq. (12) and (13): The formula looks quite arbitrary. Is there any physics behind these polynomial parametrizations or are they only empirical? Note that there are some well-established 2-stream or 4-stream formula for layer reflection, e.g., Coakley (1975). Is it possible to draw some theoretical basis or physical meaning for the parameterization from these 2-stream or 4-stream approximations? Also, some previous studies have tried to use the concept of adding doubling to approximate the reflection of two layers (Lenoble 1985). Do you think these formulae might be helpful?

Thank you for this comment. Please see our general response above, in addition to our subsequent response. While the 2-stream (or 4-stream) formulae are appropriate for many applications, they are not suited for what we capture with our parameterization (see also general comments at the beginning of this response). What we have developed in this manuscript is not about the approximation, or creating a new approximation, but to directly represent the relationship between narrowband to broadband for which the spectral dependencies are implicit. The approximation formulas presented in Coakley (1975) would not work for our application because they would still require *spectral* properties which would need to be averaged or parameterized first, which is not as direct as our parameterization.

In addition, the approximations are not sufficiently accurate for our purposes since simplifications are required in these analytical formulae. For example, the thin layer approximation by Coakley '75, eq 23 (which does take absorption into account, for non-black surfaces) starts deviating even for fairly small optical thickness as can be seen in the figure below. Additionally, the non-linearity of DARE as a function of AOD (known from RTM calculations) would not be represented by the thin layer approximation.

[Figure]

Why is SZA dependence of DARE omitted in Eq. (12) and (13)?

The dependence of DARE on SZA is not included in these equations since the parameterization coefficients themselves do not depend on SZA. Rather, the parameterization coefficients are calculated separately for a range of SZAs.

Eq. (14) and (15): again, these parameterizations look arbitrary. Is there any underlying physics?

The underlying physics are only included insofar as that we parameterized broadband DARE as a functional dependence with respect to its driving parameters. The only way to transition from narrowband to broadband is through polynomial fitting; there is no direct way to include the physics of radiative transfer in aerosol layers above clouds, which has been well established since the mid-1970s. What we have developed in this manuscript is not about the approximation, or creating a new approximation, but to directly represent the relationship between narrowband to broadband for which the spectral dependencies are implicit. The analytical formulas which are based on physics do not work for our application because they would still require *spectral* properties which would need to be averaged or parameterized first, which is not as direct as our parameterization.

Eq. 21 – 24: I understand that dSSA term is introduced to make the parameterization scheme more general and more accurate. But as I mentioned above, a broadband RTM can easily compute the DARE given any type of SSA and g. Why bother developing such a complicated parameterization?

The parameterization removes the necessity of *spectral* SSA and g which are often difficult to obtain. We fully agree with the reviewer though: An RTM model would be more appropriate to compute the DARE *if* (!) spectral properties are known. However, most often spectral quantities are not known, unless they are tied to a fixed aerosol "type" that tabulates the spectral dependence (OPAC). More importantly, we show that *we do even not need to know these spectral detail*s as *explicit* inputs since just a few (2 or 3) mid-visible parameters explain the variability of DARE across the ORACLES cases we analyzed *to within the measurement uncertainty*. The spectral dependence of the parameters as measured/retrieved is thereby *implicitly* accounted for. This will be carefully included in the revised manuscript as we apparently failed to get this crucial point across. More discussion on this point is provided in the general response at the beginning of this response.

---

## Author Comment (AC3) · 13 Nov 2020

Dear Jim Haywood,

Thank you for your positive review of our manuscript. We appreciate the comments and suggestions you provided and we believe they will greatly improve the clarity and scientific robustness of the manuscript. Below we have provided responses (in boldface) to each of the major and minor comments. Line numbers refer to the revised manuscript.

Note: the color coding for the edit markings in the document are as follows: Blue: directly addresses a comment from Reviewer #1 (you) Green: directly addresses a comment from Reviewer #2 (anonymous) Red: additional update made by Authors

Review of Cochrane et al, AMT

This paper builds on the previous work of Cochrane et al. (2019) and examines the direct radiative effect of biomass burning aerosols over cloudy and clear-sky conditions during the ORACLES 2016 and 2017 measurement campaigns. There are three main aspects: 1) assessing the aerosol optical parameters that give consistent radiative closure, 2) developing a basic and extended algorithm that parameterizes the DARE as a function of the scene albedo and the optical depth (at 550 nm) and 3) adding an additional dependency on SSA.

I was very interested and impressed by the SSA retrievals when compared to some of the more accurate assessments of SSA that are now possible using more advanced instrumentation (e.g. airborne CRD and PAS measurements that are not subject to the artefacts associated with missing scattering (nephelometers) or with scattering/absorption artefacts (filter based absorption measurements). I have included some additional references to these measurements as they were not included by the authors – I do think that these complementary studies provide excellent additional supporting information for the validity of the approach.

The second part of the paper which documents the performance of the P\_DARE and PX\_DARE could certainly be of use when compared to satellite data, but one would have to have an estimate of the above cloud/above surface AOD and the scene albedo. The most obvious place where this could have applicability might therefore be a combination of e.g. space borne lidar and broadband scene albedo from e.g. GERB; it might be worth explicitly stating this as a future possibility.

I deal with the comments as more major and minor below.

**More Major:**

Point 1: L325 onwards: It is worth emphasizing that many of the in-situ retrievals of SSA have, in the past and in the ORACLES measurements, relied on filter based measurements for

absorption and nephelometers for scattering (authors' Figure 4). These instruments have relatively large uncertainties associated with them because of corrections needed for scattering and absorbing artefacts (e.g. Cappa et al., 2008). Much more accurate measurement systems such as Cavity Ring Down (extinction) and Photo Acoustic Spectrometry have been developed.

Cappa, C. D., Lack, D. A., Burkholder, J. B., and Ravishankara, A. R.: Bias in filter-based aerosol light absorption measurements due to organic aerosol loading: Evidence from laboratory measurements, Aerosol Sci. Technol., 42, 1022–1032, https://doi.org/10.1080/02786820802389285, 2008.

I know that it is difficult to keep up with the contemporary literature, particularly with the concentrated efforts over the SE Atlantic region, but there is some recent work from the CLARIFY- 2017 team that very much supports the values of the SSA (and the wavelength dependence). I would suggest adding something like this:-

"New, more accurate, cavity ring down and photo acoustic spectrometry instrumentation has recently been deployed to the SE Atlantic during the CLARIFY-2017 deployment. Davies *et al.* (2019) performed an analysis of the SSA of aerosol dominated by biomass burning aerosol using such instrumentation and found mean SSA values of 0.84, 0.83 and 0.81 at interpolated wavelengths of 467, 528, and 652 nm respectively. Wu *et al.* (2020a) extended this analysis by examining the BBA in the free troposphere, finding a mean and variability in BBA SSA of 0.85 ± 0.02 (1stdev) and 0.82±0.04 (1stdev) at 405 and 658 nm with evidence that the BBA at higher altitudes in the free troposphere is less absorbing. These results appear entirely consistent with those derived here."

The authors should consider including a combination of the Davies et al. (2019)/Wu et al. (2020) paper on their Figure 4 as the agreement is so good.....Of course, you would have to caveat this with the fact that there are different temporal and geographical sampling regions etc.

Thank you for these comments. We apologize for our unintentional exclusion of the other SEA campaigns beyond ORACLES. We have included the suggested text on line 374. In figure 4, we have added an additional indicator of the Davies et al., 2019 results. We choose not to include the Wu et al., 2020 results since the wavelengths are not as similar.

Point 2: Line 234. The RT calculations themselves have a degree of uncertainty associated with them. For example, I note that the aerosol is characterized by the asymmetry factor. Does this mean that the higher order moments of the phase function are not accounted for? Is the RT code 2-stream? Is delta-Eddington rescaling applied? How is the surface reflectance modelled? A few more details would be appropriate here as would some acknowledgement that radiative transfer models that treat aerosols have their own inherent uncertainty (e.g. Boucher et al., 1999).

Thank you for pointing out the lack of detail surrounding the RT calculations within the manuscript. The RT calculations are performed with a multi stream RT model, disort (run with 6 streams). There is no delta-Eddington rescaling. The model assumes a Henyey-Greenstein (HG) phase function with our retrieved asymmetry parameter as input for the DARE calculations. We do not at all claim here that HG is, in fact, the actual phase function. On the contrary, the real phase function most likely deviates from HG significantly. However, since we retrieve g from *irradiance* observations, using disort, and then we use this same disort to calculate DARE, HG(g) *represents* the phase function sufficiently well for our purposes. As you know, the first moment of the phase function along with a parametric HG phase function is sufficient to calculate fluxes with a two-stream approximation, whereas higher moments are required when using higher-stream RT (including disort). The higher moments of the phase function can be generated from g (higher powers of it; for example, the second moment is simply g2). The retrieval uncertainty in g (provided in the manuscript) could be propagated into higher moments, but since they are derived from g for HG, this is not necessary here.

For the aerosol retrievals and DARE RT calculations, we simply set the "surface" to the level of the cloud and define the albedo as the spectral cloud albedo. Therefore, we do not model the surface in the main calculations within the manuscript. However, for the DARE parameterization, the spectral cloud albedo "grid" at each SZA was obtained through cloud optical thickness and effective radius retrievals.

The translation from the mid-visible albedo to the spectral albedo (described in Appendix A.3.2) starts with the originally measured albedo spectrum. From that, COD and REF are retrieved, from which the spectrum is calculated. For the other albedos in the "grid", the COD

is simply varied such that the albedo at the mid-visible wavelength changes as needed (while changing all of the other wavelengths as well). The COT/REF retrieval uses two wavelengths (860 and 1630 nm) to construct the lookup table, with no aerosol since the retrieval is done on the below aerosol, above cloud measurements. In these calculations, the surface for the cloud retrievals is standard Lambertian with an albedo value of 0.03. The COT range begins at 0, and this translates to a 0 "surface" albedo for the parameterization.

To address this comment in the manuscript, we added the following information to this section in the paper:

Lines 252/253: The RTM is run with 6 streams, assumes a Henyey Greenstein Phase function, and no delta-Eddington scaling is applied, all of which contribute to the inherent uncertainty within the RTM (Boucher et al., 1999).

Line 262: the RTM ingests the spectral cloud top albedo from SSFR (set as the surface within the model at the measured altitude, around 2km)

We should add that these simplifications were made to combine as many legs (cases) as possible in a common framework. In terms of radiative transfer, an opaque cloud effectively replaces the surface as boundary condition. However, for partially transparent clouds, the surface contributes to the reflectance. On the other hand, DARE is the *difference* between the fluxes with/without the aerosol above the altitude at which the albedo is prescribed in the model, and the approximations we used should therefore be insignificant compared to other uncertainties. Finally, it is acknowledged that clouds do not exhibit a Lambertian albedo. However, for irradiance calculations, the cloud albedo (non-Lambertian) can be substituted with a Lambertian albedo. A brief clarification on these subtleties has been included in the revised manuscript in Appendix section A.3.2, line 859-866.

Point 3: The authors chose a SZA of 20degrees to demonstrate the RT calculations and the parameterization fits. As demonstration purposes, this is OK, but there should be an acknowledgement that, when comparing to models, the DRE is typically calculated over the full range of SZAs that are experienced in the region and then diurnally averaged.

Thank you for bringing this up. To make it clear that our parameterizations are instantaneous, the sentence preceding eq 3 and eq 4 has been changed from "ORACLES measurements are used collectively to develop two parameterizations in the form of:" to "ORACLES measurements are used collectively to develop two parameterizations of instantaneous DARE in the form of:" In addition, we added the caveat to the caption figure 5: "It should be noted that a 20° SZA is not representative of the mean in the region." We have further added (lines 522-528) that in the first application of our parameterization, the diurnal integration will be done (see also our response regarding applications below).

**Minor typos/clarifications**

L24: spanned -> determined

**The abstract has been rewritten (below, in blue) to better reflect the main goals of the paper. This wording no longer remains.**

Revised Abstract: In this manuscript, we use observations from the NASA ORACLES (ObseRvations of CLouds above Aerosols and their intEractionS) aircraft campaign to develop a framework by way of two parameterizations that establishes regionally representative relationships between aerosol-cloud properties and their radiative effects. These relationships rely on new spectral aerosol property retrievals of the single scattering albedo (SSA) and asymmetry parameter (ASY). The retrievals capture the natural variability of the study region as sampled, and both were found to be fairly narrowly constrained (SSA:  $0.83 \pm 0.03$  in the mid-visible, 532 nm; ASY:  $0.54 \pm 0.06$  at 532 nm). The spectral retrievals are well suited to calculate the direct aerosol radiative effect (DARE) since SSA and ASY are tied directly to the irradiance measured in presence of aerosols – one of the inputs to the spectral DARE.

The framework allows for entire campaigns to be generalized into a set of parameterizations. For a range of solar zenith angles, it links the broadband DARE to the mid-visible aerosol optical thickness (AOD) and the albedo  $\alpha$  of the underlying scene (either clouds or clear sky) by way of the first parameterization: P(AOD,  $\alpha$ ). For ORACLES, the majority of the case-to-case variability of the broadband DARE is attributable to the dependence on the two driving parameters of P(AOD,  $\alpha$ ). A second, extended, parameterization PX(AOD,  $\alpha$ , SSA) explains even more of the case-to-case variability by introducing the mid-visible SSA  $\varpi$  as third parameter. These parameterizations establish a direct link from two or three mid-visible (narrowband) parameters to the broadband DARE, implicitly accounting for the underlying spectral dependencies of its drivers. They circumvent some of the assumptions when calculating DARE from satellite products, or in a modeling context. For example, the DARE dependence on aerosol microphysical properties is not explicit in P or PX because the asymmetry parameter varies too little from case to case to translate into appreciable DARE variability. While these particular DARE parameterizations only represent the ORACLES data, they raise the prospect of generalizing the framework to other regions.

L38: just off -> off (as the Sc extends 1000km....)

**This has been adjusted.**

L48-49: "In a region like the southeast Atlantic, this makes determining DARE challenging since the cloud fields change rapidly". I would suggest adding some idea of why the cloud changes rapidly - not only because of cloud dynamics, but because the cloud field advection is dominated by the flow in the MBL while the aerosol advection is dominated by the flow in the residual continental marine boundary layer which is frequently in the opposite direction.

**Thank you, additional details have been included here (line 53)**

L52: Chand was not the first to coin the phrase critical surface albedo. I'd suggest adding Haywood and Shine, 1995 reference here (Haywood, J.M., and Shine, K.P., 1995. The effect of anthropogenic sulfate and soot aerosol on the clear sky planetary radiation budget. Geophys. Res. Letts., 22, 5, 603- 606; see their Fig 1).

**Reference has been added. Thank you for pointing out our oversight.**

L75 (and probably other instances) aerosol optical depth -> AOD as you've already defined it

**All instances have been updated.**

I like Figure 1. It captures the essence of the filters and the criteria.

**Thank you.**

L148: Figure 3a -> Figure 2a.

**Figure reference has been updated.**

Caption for Figure 2. "c) The ratio between the BOL and TOL albedo spectra shown against the BOL AOD spectrum." I think that this needs a little more explanation. Presumably the AOD is 532nm? What about the albedo – is this the broadband albedo (i.e. weighted by the solar flux)?

Figure 2c shows the ratio between the two albedo spectra from 2b (BOL and TOL SSFR spectral albedo) as a function of the 4STAR AOD spectrum at the BOL. To make this more clear, the figure caption has been updated to: "c) The ratio between the BOL and TOL albedo spectra (taken from Fig 2b) shown against the BOL AOD spectrum at the 4STAR wavelengths. The intercept of the fit line is criteria 3 ( $AR_{\infty}$ ); if the intercept deviates largely from 1.0, the case cannot be used for an aerosol retrieval. Select wavelengths are labelled to highlight the spectral importance of this method."

In addition, the figure has been updated to highlight a few wavelengths so it is clear that the data shown is spectral. Below is the revised figure:

L163-L164: "The filter, which is applied to the upwelling profile, retains only those data within the 68% confidence interval (1 sigma) of the linear fit line". This is fine if the error distribution is Gaussian, but is it? It would be worth checking that this is the case. It seems as though the two

cases where the original filtering method is retained may not be Gaussian which might give you a reason for applying a different filtering method.

Technically, the measurement data should be Gaussian, since Gaussian is essentially measurement repeatability with some variability. For the stationary condition, if we are measuring over the same surface the measurements should be Gaussian distributed, or if noise is present, that should be Gaussian distributed. However, if the cloud changes within the measurement period, one no longer measures the same thing, and that will be reflected in the data. The idea is that through the filtering, we get rid of anything that falls outside of that Gaussian envelope since the data become too unlikely.

In the original filtering method (from Cochrane et al., 2019), the 1-sigma standard deviation limit is based on the mean value of the profile. The updated filtering method uses the 1-sigma standard deviation based on the *linear fit* of the profile. In most cases, the new filtering provided better outcomes for the quality criteria (C1, C2, C3 in Fig 1) that follow the filtering, indicating that the new filter was eliminating more of the outlying data and retaining the quality data. However, in the exception cases, we found the opposite, and the new filter resulted in poorer values of the quality criteria. In these cases, we reverted back to the old filter method.

Below are the comparisons between the new and old filtering below (shown for the two exception cases that pass the criteria for a retrieval and one for a standard case that uses the new filtering). The cyan color is data within the 1 sigma fit and therefore retained; the lighter colors (pink, orange, purple) are data that do not pass the filter and not retained. In the first case, the new filter eliminates the highest AOD data. In the second case, the new filter retains too much variability at the highest AOD (lowest altitude).

---

## Author Comment (AC4) · 13 Nov 2020

**Dear reviewer,**

Thank you for your thoughtful and constructive review of the manuscript. In the following paragraphs, we have summarized and responded to what we understand to be the main points of concern. We have also provided specific responses including plots to the individual minor comments.

Note: the color coding for the edit markings in the document are as follows: Blue: directly addresses a comment from Reviewer #1 Green: directly addresses a comment from Reviewer #2 Red: additional update made by Authors

**General comments/approach**

It is evident from your comments that we are lacking clarity in the motivation of developing a DARE parameterization, which is key to the success of the paper. We appreciate this comment and acknowledge that the purpose was not made clear throughout the manuscript. The major advantage of developing the DARE parameterization is to encapsulate the transition from narrowband to broadband such that spectral aerosol and cloud properties are not required. The parameterizations require only one wavelength of input parameters and provides output of broadband DARE. Our goal was not to create an alternative to wellestablished radiative transfer formulae such as those presented in Coakley 1975, nor to replace full RT models such as libRadtran (with its solver "disort" created by Warren Wiscombe) [See our changes/additions on lines 99-101]. Rather, the difficulty concerns the spectral aerosol and cloud properties and their relationship to the *broadband* DARE. One cannot simply translate SSA, g, cloud albedo, etc. at a single wavelength into broadband DARE since their spectral dependencies are almost as important as the narrow-band single-wavelength values. Established formulae that are grounded in the physics (such as Coakley's or other 2-stream approximations) still require spectral properties and subsequent broadband integration. Here, we replace the need for a full spectrum with narrow-band quantities at a wavelength in the mid-visible wavelength range where they are often available, and we also side-step the spectral integration. This, rather than the radiative transfer itself, is the essence of this work. More generally, we cast the collective radiative observations into an ORACLES-wide set of formulae and "resolve" the case-to-case DARE variability in terms of only two (or three) driving parameters. To express this better, we re-wrote the abstract of the paper (included in blue font below), and made changes in other places of the manuscript (e.g., lines 99-101).

Revised Abstract: In this manuscript, we use observations from the NASA ORACLES (ObseRvations of CLouds above Aerosols and their intEractionS) aircraft campaign to develop a framework by way of two parameterizations that establishes regionally representative relationships between aerosol-cloud properties and their radiative effects. These relationships rely on new spectral aerosol property retrievals of the single scattering albedo (SSA) and asymmetry parameter (ASY). The retrievals capture the natural variability of the study region as sampled, and both were found to be fairly narrowly constrained (SSA:  $0.83 \pm 0.03$  in the mid-visible, 532 nm; ASY:  $0.54 \pm 0.06$  at 532 nm). The spectral retrievals are well suited to calculate the direct aerosol radiative effect (DARE) since SSA and ASY are tied directly to the irradiance measured in presence of aerosols – one of the inputs to the spectral DARE.

The framework allows for entire campaigns to be generalized into a set of parameterizations. For a range of solar zenith angles, it links the broadband DARE to the mid-visible aerosol optical thickness (AOD) and the albedo  $\alpha$  of the underlying scene (either clouds or clear sky) by way of the first parameterization: P(AOD,  $\alpha$ ). For ORACLES, the majority of the case-to-case variability of the broadband DARE is attributable to the dependence on the two driving parameters of P(AOD,  $\alpha$ ). A second, extended, parameterization PX(AOD,  $\alpha$ , SSA) explains even more of the case-to-case variability by introducing the mid-visible SSA  $\varpi$  as third parameter. These parameterizations establish a direct link from two or three mid-visible (narrowband) parameters to the broadband DARE, implicitly accounting for the underlying spectral dependencies of its drivers. They circumvent some of the assumptions when calculating DARE from satellite products, or in a modeling context. For example, the DARE dependence on aerosol microphysical properties is not explicit in P or PX because the asymmetry parameter varies too little from case to case to translate into appreciable DARE variability. While these particular DARE parameterizations only represent the ORACLES data, they raise the prospect of generalizing the framework to other regions.

The *application* of the parameterizations has been made clearer in the revised manuscript, with the important addition of explicitly stating how this is already being used to obtain diurnally-integrated values of DARE for the study region (lines 514-519). We are actively working on a publication that uses the parameterization to establish diurnally-integrated DARE for the study region, based on satellite observations and aircraft observations of AOD and albedo. It is important to note, however, that the parameterizations were not developed for regions beyond that of ORACLES, since as you pointed out, the physics in other regions are most likely different and ours is an empirical formula that applies only to the data it is based on (we alluded to aerosol microphysics specifically in the revised abstract quoted above). While we recognize that the number of cases is small, the range of properties they cover is likely sufficiently representative of the conditions in the study region for developing the parameterization scheme. As more cases become available, even those falling outside of the envelope of, say, previously observed single scattering albedo, the relationship between midvisible aerosol properties (AOD, SSA, asymmetry parameter), albedo on the one hand, and broadband DARE on the other would likely be similar as well. The advantage of our framework is that it can be continually updated. This will be done, for example, with the aforementioned paper that we are currently working on. Whereas the present paper only considers ORACLES 2016 and 2017, the follow-up paper extends the parameterization to ORACLES 2018, prior to the application of the parameterization to the calculation of campaign-averaged DARE. Data from other campaigns (e.g., from CLARIFY or even SAFARI) could similarly be included.

We understand that the two parts of the paper seem distinct from one another, and this is done purposefully. The manuscript is the second part to its companion paper, Cochrane et al., 2019, and they are meant to be read in succession. However, it was important that we include enough detail in the first part of the paper on the retrieval algorithm such that it can be understood at a surface level without reading the 2019 companion paper, as well as to make clear the changes made to the algorithm relative to the version described in the 2019 paper. The most important change of the 2020 version of the algorithm relative to its 2019 predecessor is that the data are automatically filtered and conditioned in a consistent manner. This allows ingesting of data without different treatment of cases and thereby biasing the results. In addition, part I of the current paper provides a data product on its own: the aerosol property retrievals of SSA and g. This database of these retrievals, though small, is important to include along with the parameterization of DARE as presented in part II.

To address your concerns, here is what we have done within the revised manuscript:

First, the abstract has been revised to better reflect the overall goals of the paper (see draft above in blue). Second, the two parts of the paper are now delineated at the end of the introduction (Lines 115-119). Regarding the body of the paper: The original version did not effectively capture the main objectives of the paper and potentially left the reader with an incorrect impression of what the paper is about. In the revised version, the benefit of a direct transition from narrowband to broadband through our parameterization has been highlighted in greater detail (lines 309-314). In addition, the application of our parameterizations has been more explicitly described (using the next paper as example; lines 514-519). We have made the separation between part I and part II of the paper even more apparent, making it possible to skip the aerosol property retrieval section and move directly into the DARE section (lines 115-119; section headings). However, we are hesitant to significantly reduce this section because of its value to the southeast Atlantic region research community.

Thank you again for your review of the paper and we hope that our changes have addressed your concerns satisfactorily.

With the above comments, we believe that we addressed the general (major) comments. Our response to sequential (minor) comments is below (in blue).

Minor comments:

Line 24: Does the "scene albedo" actually mean cloud albedo? If it really means "scene albedo" then what types of scenes (ocean, land, snow etc.) have been included?

ORACLES data was exclusively taken over the southeast Atlantic ocean. We use "scene albedo" rather than "cloud albedo" since we encountered scenes that included both cloudy and clear sky (partially cloudy or area contained cloud holes.) This is clarified in the revised manuscript at line 92 (and described in the newly drafted abstract quoted above).

Line 86: similarly, it should be clarified here if "scene albedo" actually just means "cloud albedo". Note that the spectral signature of land reflectance/albedo is very different from cloud albedo.

Thank you for pointing out this ambiguity. For more clarity, line 92 has been changed from "The 550 nm albedo is the albedo of the scene below the aerosol layer and the SSA is a measure of aerosol absorption." to "The 550 nm albedo is the albedo of the scene below the aerosol layer (open ocean and/or cloudy scene), and the SSA is a measure of aerosol absorption." A detailed discussion of the albedo has also been included in Appendix A.3.2.

Eq. (3) and (4): It should be pointed out explicitly if these equations are for instantaneous or diurnal averaged DARE. Also why is the dependence of DARE on solar zenith angle omitted in these equations? SZA is part of the parameterization (Table 4b), no?

Thank you, the sentence preceding eq 3 and eq 4 (line 85) has been changed from "ORACLES measurements are used collectively to develop two parameterizations in the form of:" to "ORACLES measurements are used collectively to develop two parameterizations of instantaneous DARE in the form of:"

The dependence of DARE on SZA is not included in these equations, since the parameterization coefficients themselves do not depend on SZA. Rather, the parameterization coefficients are calculated separately for a range of SZAs.

To what extent is the DARE dependent on atmospheric profiles, such as water vapor profiles? There are some recent studies that suggest a correlation between the presence of above-cloud smoke and an enhanced water vapor in the ORACLE region. Should this correlation be considered in the parameterization?

This is an important question. The dependence of DARE on atmospheric profiles, specifically water vapor, is implicitly included within the parameterization. Each case that the parameterization is based upon has the measured water vapor profile included in its DARE calculations.

While we do not determine the extent to which DARE depends on water vapor, it should not depend too much on it since DARE is defined as Fnet (with aerosol) – Fnet (without aerosol). The water vapor profile for both terms was held fixed (as determined from observations for each of the cases), and therefore only has a bearing on DARE if it affects the incident radiation on top of the aerosol layer. Compared to the impact of the scene AOD, albedo, and SSA on the case-to-case DARE variability, the water vapor profile variability (as well as the aerosol microphysics variability) was negligible. However, the water vapor does have a bearing on the total heating rates, which is explored in a separate paper (in progress; if submitted before this paper is accepted, we will include a reference in the appropriate place)

Cochrane et al., 2019.has been cited many times in the paper, often in different formats. Please be consistent and also considering use abbreviation e.g., C19 to refer to Cochrane et al., 2019.

Thank you for catching this. The references to Cochrane et al., 2019 have been abbreviated to C19 (defined upon its first occurrence).

Line 148: Figure 3 should be Figure 2.

Thank you. This has been updated in line 156.

Eq. (5) and (6), why are the parameters a\_lambda and b\_lambda the same for upward and downward irradiances? What is the underlying physics?

Thank you for pointing this out, as our notation is ambiguous.  $a_{\lambda}^{\uparrow}$  is not the same as  $a_{\lambda}^{\downarrow}$  (similar for  $b_{\lambda}^{\uparrow}$  and  $b_{\lambda}^{\downarrow}$ ; equation 6 has been changed to  $c_{\lambda}^{\uparrow}$  and  $d_{\lambda}^{\downarrow}$  (line 171).

Again, Section 2.2.1 and 2.2.2 seem to be a replay of Cochrane et al., 2019. They should be put in the Appendix or substantially shortened.

As noted in the response to the general comments, we understand that the two parts of the paper seem distinct from one another, and this is done purposefully. The manuscript is the second part to its companion paper, Cochrane et al., 2019, and they are meant to be read in succession. However, it is important that we include enough detail in the first part of the paper on the retrieval algorithm such that it can be understood at a surface level without reading the 2019 companion paper, as well as to make clear the changes made to the algorithm relative to the version described in the 2019 paper. The most important change of the 2020 version of the algorithm relative to its 2019 predecessor is that the data are automatically filtered and conditioned in a consistent manner. This allows ingesting of data without different treatment of cases and thereby biasing the results. In addition, part I of the current paper provides a data product on its own: the aerosol property retrievals of SSA and g. This database of these retrievals, though small, is important to include along with the parameterization of DARE as presented in part II.

To address your concern, we have added additional text on lines 115-119 to better describe the two parts of the paper and have updated the section headings to reflect the separation (line 326).

Around line 150, this part is confusing and needs detail explanation. For example, "both upwelling (F\_up) and downwelling (F\_dn) irradiance profiles have an approximately linear relationship to AOD due to the absorption and scattering of the aerosol layer." Shouldn't the downwelling (F\_dn) be exponential with AOD as a result of Beer's law? "Any deviation from the linear relationship is attributed to changes in the underlying cloud" Why? Can't the vertical variation of aerosol properties, e.g., SSA and/or g cause deviation from the linear relationship? How does cloud cause the deviation? These questions need to be clarified.

We did not intend to imply that the direct downwelling irradiance should not follow Beer's law. However, in the SSFR measurements we cannot separate the global irradiance into its diffuse and direct components. We are using the linear assumption for the global downwelling as a simplification only for initial fitting for the subsequent filtering. We agree with the reviewer here because indeed, deviations from the linear relationship could be due to non-linearities as expected from Beer's law, or vertical dependencies of aerosol parameters. However, we expect these to be negligible compared to changes in the underlying clouds, and therefore use deviations from a linear profile – just to filter our data. Otherwise, the results of the paper (i.e., the retrievals and the calculations) are not predicted on any linearity assumptions.

As can be seen from the figures below, the direct-beam downwelling (labeled SPN-f, SPN-S) *does* follow an exponential (these measurements were only available during 2018, and they are therefore not included in this manuscript). However, the global downwelling (SSFR global, green) can be approximated with a linear fit, at least for the range of AOD we encountered. We should add that the thin-layer approximation introduced by, e.g., Coakley et al. (1975) shows a

linear dependence of the global reflectance analytically ("In the thin atmosphere approximation, the reflectivity and transmissivity of the layer are linear functions of its optical depth"). The figure below on the right shows the irradiance components as a function of altitude.

While vertical variation of the aerosol properties could theoretically affect the linear relationship that we assumed, but changes in the cloud properties below outweigh aerosol-induced deviations from a linear profile by far, and can therefore be used as the basis for filtering.

To address this comment, we have added the following additional text on lines 159-163:

"This linear assumption for the global downwelling is a simplification only for initial fitting for the subsequent filtering, and deviations from the linear relationship could be due to nonlinearities as expected from Beer's law, or vertical dependencies of aerosol parameters. However, we expect these to be negligible compared to changes in the underlying clouds, and therefore use deviations from a linear profile to filter our data."

Does the retrieval algorithm assume H-G phase function? What are the higher-order terms of the phase function expansion other than asymmetry factor, g? What is the uncertainty associated with the phase function assumption?

Yes, it does assume an Henyey-Greenstein (HG) phase function with our retrieved asymmetry parameter (the first moment of the actual phase function) as input. The first moment of the phase function along with a parametric HG phase function is sufficient to calculate fluxes with a two-stream approximation. Higher moments are required when using higher-stream RT models (including disort) and can be generated from g (higher powers of it; for example, the second moment is simply  $g^2$ ). We do not at all claim here that HG is, in fact, the actual phase function. On the contrary, the real phase function most likely deviates from HG. However, HG(g) *represents* the real phase function sufficiently well for our purposes. Using Coakley '75 again as an example: He does show that there are differences in the reflectance when using a two-stream formula relative to a multi-stream (in this case, doubling adding) RT. However, these differences arose because they were looking at directional reflectance. In our case, we retrieve g from *irradiance* observations, using disort, and then we use this same disort to calculate DARE. The retrieval uncertainty in g (provided in the manuscript) could propagate into higher moments, but since they are trivially derived from g for HG ( $g^2$ ,  $g^3$ , etc.), this is not necessary here.

The following additional text is included at lines 253/254:

"The RTM is run with 6 streams, assumes a Henyey Greenstein Phase function, and no delta-Eddington scaling is applied, all of which contribute to the inherent uncertainty within the RTM (Boucher et al., 1999)."

Line 286: Russel et al., 1997 should be Russell et al., 1997; deGraaf should be de Graaf

**Thank you for pointing out these errors. We have removed this statement and the associated references due to a comment from another reviewer.**

Eq. (12) and (13): The formula looks quite arbitrary. Is there any physics behind these polynomial parametrizations or are they only empirical? Note that there are some well-established 2-stream or 4-stream formula for layer reflection, e.g., Coakley (1975). Is it possible to draw some theoretical basis or physical meaning for the parameterization from these 2-stream or 4-stream approximations? Also, some previous studies have tried to use the concept of adding doubling to approximate the reflection of two layers (Lenoble 1985). Do you think these formulae might be helpful?

Thank you for this comment. Please see our general response above, in addition to our subsequent response. While the 2-stream (or 4-stream) formulae are appropriate for many applications, they are not suited for what we capture with our parameterization (see also general comments at the beginning of this response). What we have developed in this manuscript is not about the approximation, or creating a new approximation, but to directly represent the relationship between narrowband to broadband for which the spectral dependencies are implicit. The approximation formulas presented in Coakley (1975) would not work for our application because they would still require *spectral* properties which would need to be averaged or parameterized first, which is not as direct as our parameterization.

In addition, the approximations are not sufficiently accurate for our purposes since simplifications are required in these analytical formulae. For example, the thin layer approximation by Coakley '75, eq 23 (which does take absorption into account, for non-black surfaces) starts deviating even for fairly small optical thickness as can be seen in the figure below. Additionally, the non-linearity of DARE as a function of AOD (known from RTM calculations) would not be represented by the thin layer approximation.

To address your comment, we have added the following additional text on lines 99-101: "They are not meant to replace detailed or approximated radiative transfer calculations (e.g., Coakley 1975), which would require all these inputs, but rather to arrive at a broadband DARE with a minimum set of input parameters that drive its regional variability."

Why is SZA dependence of DARE omitted in Eq. (12) and (13)?

The dependence of DARE on SZA is not included in these equations since the parameterization coefficients themselves do not depend on SZA. Rather, the parameterization coefficients are calculated separately for a range of SZAs.

Eq. (14) and (15): again, these parameterizations look arbitrary. Is there any underlying physics?

The underlying physics are only included insofar as that we parameterized broadband DARE as a functional dependence with respect to its driving parameters. The only way to transition from narrowband to broadband is through polynomial fitting; there is no direct way to include the physics of radiative transfer in aerosol layers above clouds, which has been well established since the mid-1970s. Of course, one could calculate the broadband DARE if all of the parameters were available, but that is not the point of this paper (see the other comments above). What we have developed in this manuscript is not about the approximation, or creating a new approximation, but to directly represent the relationship between narrowband to broadband for which the spectral dependencies are implicit. The analytical formulas which are based on physics do not work for our application because they would still require *spectral* properties which would need to be averaged or parameterized first, which is not as direct as our parameterization.

To make this more clear in the manuscript, additional text has been added on lines 99-101 and 309-314;

Eq. 21 - 24: I understand that dSSA term is introduced to make the parameterization scheme more general and more accurate. But as I mentioned above, a broadband RTM can easily compute the DARE given any type of SSA and g. Why bother developing such a complicated parameterization?

The parameterization removes the necessity of *spectral* SSA and g which are often difficult to obtain. We fully agree with the reviewer though: An RTM model would be more appropriate to compute the DARE *if* (!) spectral properties were known. However, most often spectral quantities are *not* known, unless they are tied to a fixed aerosol "type" that tabulates the spectral dependence (e.g., OPAC). More importantly, we show that *we do even not need to know these spectral details* as *explicit* inputs since just a few (2 or 3) mid-visible parameters explain the variability of DARE across the ORACLES cases we analyzed *to within the measurement uncertainty*. The spectral dependence of the parameters as measured/retrieved is thereby *implicitly* accounted for, and we can thus obtain broadband DARE with a minimum number of input parameters.

We have addressed this comment with revised abstract as well as the additional text on lines 99-101; 309-314; 522-528.

**List of relevant changes:**

- P.1-2, L. 21-39 (revised abstract)
- P.3, L. 91-92
- P.3, L. 99-101
- P.4, L. 110
- P.4, L. 115-119
- P.5, L. 157, 163, 164
- P.5, L. 158-162
- P.6, L. 170-173
- P.6, L. 177, 182
- P.7, L. 203, 211, 216
- P.8, L. 246
- P.8, L252-253
- P.9, L. 260
- P.10, L309-314
- P.11, L. 326
- P.17, L. 522-528

In addition to edits made according to the review comments, we have made the following updates:

• Added uncertainty estimates to table 4b.

- Updated the SSA extrapolation and description (figure A1a), requiring the RT calculations be re-calculated.
- Updated Figures 5/9/10/11/D1/D2 and tables 4a and 4b to reflect the new calculations
- Included supplementary material of code and coefficient files